# Optimistic Bayesian Optimization with Unknown Constraints

**Quoc Phong Nguyen[1], Wan Theng Ruth Chew[2], Le Song[2],**
**Bryan Kian Hsiang Low[2] & Patrick Jaillet[1]**
[1]LIDS and EECS, Massachusetts Institute of Technology, USA
[2]School of Computing, National University of Singapore, Singapore
`qphongmp@gmail.com`, `chew.ruth@u.nus.edu`, `le.song@u.nus.edu`,
`lowkh@comp.nus.edu.sg`, `jaillet@mit.edu`

## Abstract

Though some research efforts have been dedicated to constrained *Bayesian optimization* (BO), there remains a notable absence of a principled approach with a theoretical performance guarantee in the *decoupled setting*. Such a setting involves independent evaluations of the objective function and constraints at different inputs, and is hence a relaxation of the commonly-studied *coupled setting* where functions must be evaluated together. As a result, the decoupled setting requires an adaptive selection between evaluating either the objective function or a constraint, in addition to selecting an input (in the coupled setting). This paper presents a novel constrained BO algorithm with a provable performance guarantee that can address the above relaxed setting. Specifically, it considers the fundamental trade-off between exploration and exploitation in constrained BO, and, interestingly, affords a noteworthy connection to active learning. The performance of our proposed algorithms is also empirically evaluated using several synthetic and real-world optimization problems.

## 1 Introduction

In real-world applications, we often encounter expensive-to-evaluate *black-box* objective functions that can only be assessed through simulations or experimentation. For example, problems involve optimizing the hyperparameters of a machine learning model (Wistuba et al., 2018; Perrone et al., 2020), or choosing experiments in the fields of material and drug design (Schweidtmann et al., 2018). To address these problems, *Bayesian optimization* (BO) has gained prominence as a widely adopted approach (Brochu et al., 2010; Frazier, 2018; Garnett, 2022). It is an iterative model-based approach that employs a probabilistic model, e.g., a *Gaussian process* (GP), to estimate the unknown objective function. At each iteration, BO searches for the optimal solution by strategically selecting an *input query* to evaluate the objective function, maintaining a balance between exploiting promising areas and exploring poorly-estimated regions. BO encompasses many well-established techniques such as the probability of improvement (Kushner, 1964), expected improvement (EI) (Mockus et al., 1978), Gaussian process upper confidence bound (GP-UCB) (Srinivas et al., 2010), the knowledge-gradient based approach (Frazier et al., 2008), and information-theoretic approaches: entropy search (Hennig and Schuler, 2012), predictive entropy search (PES) (Hernández-Lobato et al., 2014), and max-value entropy search (MES) (Wang and Jegelka, 2017).

Beyond the black-box objective function, recent advancements in BO have focused on addressing the prevalent presence of *black-box constraints*. For example, there often exist prediction time constraints and class-wise performance constraints when tuning machine learning models (Hernández-Lobato et al., 2016; Takeno et al., 2022). They are just as costly to evaluate as the objective function. Constrained BO has led to many BO extensions such as EIC (an EI-based method) (Gardner et al., 2014), a knowledge gradient-based method (Chen et al., 2021), CMES-IBO (an MES-based method) (Takeno et al., 2022), augmented Lagrangian approaches (Gramacy et al., 2016; Picheny et al., 2016), and upper trust bound (UTB) (a GP-UCB-based method) (Priem et al., 2020).

However, existing approaches primarily concentrate on empirical performance and lack a theoretical analysis to ensure consistent performance. Recently, theoretical studies for constrained BO have gained attention via the works of Lu and Paulson (2022) and Xu et al. (2023). While Lu and Paulson (2022) perform the analysis by proposing a penalty-based regret, Xu et al. (2023) analyse the cumulative regret due to the objective function and the constraint violation separately.

Besides the above largely unexplored theoretical analysis, existing works often overlook the potential for evaluating the objective function and constraints independently at different inputs, known as *decoupled queries*. Specifically, the above works, including the theoretical studies by Lu and Paulson (2022) and Xu et al. (2023), require simultaneous evaluations of the objective function and constraints at an input query, known as *coupled queries*. The distinction between coupled and decoupled queries was first mentioned in the work of Gelbart et al. (2014). They discuss a chicken-and-egg pathology which prevents extending a *myopic* BO approach such as EIC to the decoupled setting. Later, Hernández-Lobato et al. (2016) introduce a principled approach based on PES, namely PESC, to address constrained BO with decoupled queries. While PESC is functionally equivalent to a lookahead approach, it leverages the symmetric property of mutual information to avoid performing the actual lookahead computation. Regrettably, its implementation is fairly complex, making it less accessible to practitioners. Besides, it lacks a theoretical performance guarantee. While ADMMBO (Ariafar et al., 2019) deals with decoupled queries, it deterministically alternates the evaluations of the objective function and constraints, without exploiting the benefits of an adaptive selection approach. Hence, the question of devising an approach that offers a theoretical performance guarantee, is adaptable to decoupled queries, and can be readily implemented by practitioners remains unanswered.

In this paper, we address this question by proposing a simple algorithm with a theoretical performance guarantee, especially in the decoupled setting. Notably, our algorithm is myopic without expensive lookaheads. In Sec. 2, we introduce a regret that does not require any penalty parameter, unlike that in the work of Lu and Paulson (2022). Then, we discuss the exploration-exploitation trade-off in constrained BO in Sec. 3. Specifically, we introduce a new form of exploration, namely *horizontal exploration*, resulted from the presence of black-box constraints. It is to differentiate from the *vertical exploration* in unconstrained BO (Srinivas et al., 2010). Then, we design a unified approach that handles coupled and decoupled queries from this perspective. More importantly, our algorithms are shown to be no-regret in Theorem 3.3 and App. B. While the viewpoint of balancing exploration and exploitation is inherently grounded in BO, especially in the bandit setting like GP-UCB (Srinivas et al., 2010), Sec. 3.3 shows that the choice of the function to query can also be framed within the well-known uncertainty sampling paradigm in the active learning literature (Settles, 2009). In Sec. 3.4, we propose an estimator for approximating the optimal solution at each BO iteration with a theoretical performance guarantee. To empirically demonstrate the performance of our algorithms, we presents several experiments using both synthetic and real-world optimization problems in Sec. 4.

## 2 Constrained Bayesian Optimization and Regret Definition

Let $f$ be a real-valued black-box objective function and $\mathcal{C}$ be a finite set of real-valued black-box constraints. Let the compact subset $\mathcal{X} \subset \mathbb{R}^d$ be the input domain and $d \in \mathbb{N}_+$ be the input dimension. We consider the following constrained optimization problem

$$\max_{\mathbf{x} \in \mathcal{S}} f(\mathbf{x}) \text{ where the } \textit{feasible region } \mathcal{S} \triangleq \{\mathbf{x} \in \mathcal{X} \,|\, c(\mathbf{x}) \geq \lambda_c \,\forall c \in \mathcal{C}\} \text{ and } \lambda_c \in \mathbb{R} \,\forall c \in \mathcal{C} \,. \quad (1)$$

An equality constraint can be transformed into two inequality constraints. Let us denote the set of the objective function and constraints as $\mathcal{F} \triangleq \{f\} \cup \mathcal{C}$ and denote a function in $\mathcal{F}$ as $h$.

To identify the optimal solution $\mathbf{x}^* \triangleq \arg\max_{\mathbf{x} \in \mathcal{S}} f(\mathbf{x})$, we employ BO which is an algorithm operating in a sequential manner. In the *decoupled query* setting, at iteration $t$, we gather a noisy observation of the *function query* $h_t \in \mathcal{F}$ evaluated at an *input query* $\mathbf{x}_t \in \mathcal{X}$

$$y_{h_t}(\mathbf{x}_t) \triangleq h_t(\mathbf{x}_t) + \epsilon_{h_t}(\mathbf{x}_t), \quad \epsilon_{h_t}(\mathbf{x}_t) \sim \mathcal{N}(0, \sigma_{h_t}^2) \,. \quad (2)$$

For instance, at iteration $t$, the algorithm may decide to query the objective function (i.e., $h_t = f$) while it may query a constraint (i.e., $h_{t'} = c$ for some $c \in \mathcal{C}$) at a different iteration $t' \neq t$. On the contrary, in the *coupled query* setting, at iteration $t$, we query for observations $\{y_h(\mathbf{x}_t)\}_{h \in \mathcal{F}}$ of all functions $h \in \mathcal{F}$ evaluated at the same *input query* $\mathbf{x}_t \in \mathcal{X}$. The coupled setting is less challenging since it does not require specifying the function query.

Let $\mathcal{D}_{h,t}$ denote the set of observed inputs of $h$ until iteration $t$ (including $\mathbf{x}_t$) and $\mathcal{D}_t \triangleq \cup_{h \in \mathcal{F}} \mathcal{D}_{h,t}$. Then, $\mathcal{D}_0$ consists of initial observed inputs. Let $\mathbf{y}_h(\mathcal{D}_t) = \mathbf{y}_h(\mathcal{D}_{h,t}) \triangleq \{y_h(\mathbf{x})\}_{\mathbf{x} \in \mathcal{D}_{h,t}}$ denote the set of observations from $h$ at $\mathcal{D}_{h,t}$. BO effectively utilizes the acquired observations $\{\mathbf{y}_h(\mathcal{D}_{h,t-1})\}_{h \in \mathcal{F}}$ in the previous $t-1$ iterations to formulate a strategy for determining the next input query $\mathbf{x}_t$, the next function query $h_t$, and an estimator, denoted as $\tilde{\mathbf{x}}_t^*$, for approximating the optimal solution $\mathbf{x}^*$. We will discuss the probabilistic model of $h \in \mathcal{F}$ given the acquired observations and a performance metric of BO, called the *regret*, in the rest of this section. Then, we will elaborate on our strategy of choosing $\mathbf{x}_t$, $h_t$, and $\tilde{\mathbf{x}}_t^*$ in the following sections.

**Gaussian process.** For each $h \in \mathcal{F}$, we model $h$ with a *Gaussian process* (GP). It implies that every finite subset of $\{h(\mathbf{x})\}_{\mathbf{x} \in \mathcal{X}}$ follows a multivariate Gaussian distribution. The GP is fully specified by its prior mean $m_h(\mathbf{x})$ and its kernel $k_h(\mathbf{x}, \mathbf{x}') \triangleq \text{cov}(h(\mathbf{x}), h(\mathbf{x}'))$. We employ the commonly-used *squared exponential* (SE) kernel. At iteration $t$, given the observations $\mathbf{y}_h(\mathcal{D}_{t-1})$ in the previous $t-1$ iterations, the posterior distribution of $h(\mathbf{x})$ follows a Gaussian distribution with a closed-form posterior mean and variance, denoted as $\mu_{h,t-1}(\mathbf{x})$ and $\sigma_{h,t-1}^2(\mathbf{x})$, respectively.[1]

**Regrets.** To analyse the theoretical performance of constrained BO, we propose the following *instantaneous regret* $r$ including that of the objective function $r_f$ and the constraints $r_c$.

$$r(\mathbf{x}_t) \triangleq \max_{h \in \mathcal{F}} r_h(\mathbf{x}_t) \quad \text{where} \quad \begin{aligned} r_f(\mathbf{x}_t) &\triangleq \max(0, f(\mathbf{x}^*) - f(\mathbf{x}_t)) \\ \forall c \in \mathcal{C}, \ r_c(\mathbf{x}_t) &\triangleq \max(0, \lambda_c - c(\mathbf{x}_t)) \ . \end{aligned} \tag{3}$$

Then, our goal is to design BO algorithms that achieve a sublinear cumulative regret

$$\lim_{T \to \infty} \frac{1}{T} R_T \triangleq \lim_{T \to \infty} \frac{1}{T} \sum_{t=1}^{T} r(\mathbf{x}_t) = 0 \ . \tag{4}$$

as it implies that $\min_{\mathbf{x} \in \{\mathbf{x}_t\}_{t=1}^{T}} r(\mathbf{x}_t) \leq \frac{1}{T} \sum_{t=1}^{T} r(\mathbf{x}_t)$ approaches 0 as $T$ approaches $\infty$.

*Remark* 2.1 (Instantaneous regret as a sum). Alternatively, the instantaneous regret can be defined as a sum of instantaneous regrets of the objective function and the constraints

$$s(\mathbf{x}_t) \triangleq \sum_{h \in \mathcal{F}} r_h(\mathbf{x}_t) \ . \tag{5}$$

Let us consider the case of a single constraint $\mathcal{C} = \{c_0\}$. For $\mathbf{x}_t \neq \mathbf{x}_t'$, if $r_f(\mathbf{x}_t) = r_f(\mathbf{x}_t') = 1$ and $r_{c_0}(\mathbf{x}_t) = 0$ while $r_{c_0}(\mathbf{x}_t') = 1$, then $r(\mathbf{x}_t) = r(\mathbf{x}_t') = 1$ while $s(\mathbf{x}_t) = 1 < 2 = s(\mathbf{x}_t')$. As a result, $s(\mathbf{x})$ is more effective than $r(\mathbf{x})$ at measuring the suboptimality of a solution. Nevertheless, a sublinear cumulative regret w.r.t. $r$ implies a sublinear cumulative regret w.r.t. $s$ and vice versa since $r(\mathbf{x}_t) \leq s(\mathbf{x}_t) \leq |\mathcal{F}| \, r(\mathbf{x}_t)$. We revisit $s(\mathbf{x}_t)$ in Sec. 3.4 when discussing the estimator $\tilde{\mathbf{x}}_t^*$.

## 3 OPTIMISTIC BAYESIAN OPTIMIZATION WITH UNKNOWN CONSTRAINTS

To simplify the derivation, we consider the case of finite input domain $\mathcal{X}$ and utilize the following Lemma 5.1 of Srinivas et al. (2010) with a modification by applying the union bound for all functions in $\mathcal{F}$ (Lu and Paulson, 2022).

**Lemma 3.1.** *Pick $\delta \in [0, 1]$ and set $\beta_t = 2 \log(|\mathcal{F}||\mathcal{X}|t^2\pi^2/6\delta)$. Then,*

$$|h(\mathbf{x}) - \mu_{h,t-1}(\mathbf{x})| \leq \beta_t^{1/2} \sigma_{h,t-1}(\mathbf{x}) \quad \forall \mathbf{x} \in \mathcal{X} \ \forall t \geq 1 \ \forall h \in \mathcal{F} \tag{6}$$

*holds with probability $\geq 1 - \delta$. It suggests*

$$u_{h,t-1}(\mathbf{x}) \triangleq \mu_{h,t-1}(\mathbf{x}) + \beta_t^{1/2} \sigma_{h,t-1}(\mathbf{x}) \ \text{and} \ l_{h,t-1}(\mathbf{x}) \triangleq \mu_{h,t-1}(\mathbf{x}) - \beta_t^{1/2} \sigma_{h,t-1}(\mathbf{x}) \tag{7}$$

*as the* upper *and* lower *confidence bounds of $h(\mathbf{x})$ for all $h \in \mathcal{F}$, $\mathbf{x} \in \mathcal{X}$, and $t \geq 1$, respectively.*

---

[1]Please refer to Rasmussen and Williams (2006) for the closed-form expressions.

### 3.1 INPUT QUERY

In the classic unconstrained GP-UCB work of Srinivas et al. (2010) (i.e., $\mathcal{C} = \emptyset$), it balances between exploiting the current posterior belief by selecting those with high posterior mean $\mu_{f,t-1}(\mathbf{x})$, and exploring inputs with highly uncertain evaluations of $f$ by selecting those with high posterior standard deviations $\sigma_{f,t-1}(\mathbf{x})$. Specifically, GP-UCB (Srinivas et al., 2010) selects an input query that maximizes an *optimistic objective function evaluation* $u_{f,t-1}(\mathbf{x})$

$$\mathbf{x}_t^{\text{GP-UCB}} = \arg\max_{\mathbf{x}\in\mathcal{X}} u_{f,t-1}(\mathbf{x}) = \arg\max_{\mathbf{x}\in\mathcal{X}} \left( \mu_{f,t-1}(\mathbf{x}) + \underbrace{\beta_t^{1/2}\sigma_{f,t-1}(\mathbf{x})}_{\text{vertical exploration bonus}} \right). \tag{8}$$

Let us call $\beta_t^{1/2}\sigma_{f,t-1}(\mathbf{x})$ the *vertical exploration bonus* as it encourages us to be optimistic about the unknown objective function evaluations.[2] It is shown as the blue region in Fig. 1a.

In the presence of unknown constraints $\mathcal{C}$, apart from the above *optimistic objective function evaluation* $u_{f,t-1}(\mathbf{x})$, we additionally consider an *optimistic feasible region*, denoted as $\mathcal{O}_t$, which may contain some infeasible inputs. It represents the exploration of the feasible region which we refer to as *horizontal exploration* as opposed to the vertical exploration in the optimistic objective function evaluation. Ideally, $\mathcal{O}_t$ is a superset of the feasible region $\mathcal{S}$ to ensure that the optimal solution $\mathbf{x}^*$ remains in $\mathcal{O}_t$. We consider the following optimistic feasible region

$$\mathcal{O}_t \triangleq \{\mathbf{x} \in \mathcal{X} \mid u_{c,t-1}(\mathbf{x}) \geq \lambda_c \, \forall c \in \mathcal{C}\} \tag{9}$$

which aligns with our goal that $\mathcal{O}_t \supset \mathcal{S}$ with high probability since $u_{c,t-1}(\mathbf{x}) \geq c(\mathbf{x})$ with high probability (Lemma 3.1). By considering an optimistic feasible region, we avoid the subtle issue that the probability mass of the feasible region is 0 given the GP posterior beliefs of the constraints. This issue affects several existing approaches such as EIC, PESC, and CMES-IBO as discussed in App. A.

Combining the vertical and horizontal explorations, we select the input query $\mathbf{x}_t$ that maximizes the optimistic objective function $u_{f,t-1}$ restricted to the optimistic feasible region $\mathcal{O}_t$

$$\mathbf{x}_t \triangleq \arg\max_{\mathbf{x}\in\mathcal{O}_t} u_{f,t-1}(\mathbf{x}). \tag{10}$$

We derive an upper confidence bound of $r_f(\mathbf{x}_t)$ similar to that in GP-UCB, and an additional upper confidence bound of $r_c(\mathbf{x}_t)$ which holds with probability $\geq 1 - \delta$ (see App. B)

$$r_f(\mathbf{x}_t) \leq 2\beta_t^{1/2}\sigma_{f,t-1}(\mathbf{x}_t) \quad \text{and} \quad r_c(\mathbf{x}_t) \leq 2\beta_t^{1/2}\sigma_{c,t-1}(\mathbf{x}_t). \tag{11}$$

If the queries are *coupled*, we can achieve a no-regret BO algorithm by simply obtaining observations $\{y_h(\mathbf{x}_t)\}_{h\in\mathcal{F}}$ at iteration $t$ as elaborated in App. B. This algorithm for the coupled setting is called UCB-C to distinguish it from another algorithm, namely UCB-D, for the decoupled setting described in the next section.

While a condition resembling equation (9) is utilized in the work of Priem et al. (2020), they do not offer any theoretical analysis. Furthermore, they maximize a variant of the EI criterion as opposed to the upper confidence bound $u_{f,t-1}$ in our approach. Besides, the optimization problem in equation (10) can also be framed as the unconstrained penalized acquisition function in the work of Lu and Paulson (2022). It suffers from an extra penalty parameter requiring automated fine-tuning, which is left as a future work by Lu and Paulson (2022). The algorithm most closely related to our UCB-C is the recent CONFIG algorithm (Xu et al., 2023), which is accompanied by theoretical bounds on the cumulative regret from the objective function and the constraints. However, decoupled queries remain unexplored in these studies. In the next section, we address this scenario by adaptively selecting the *function query* $h_t$ while maintaining the theoretical performance guarantee.

In order for the choice of $\mathbf{x}_t$ in equation (10) to exist, $\mathcal{O}_t$ must be non-empty. This holds with probability $\geq 1 - \delta$ according to Lemma 3.1 if the optimization problem is feasible. To avoid the subtle case of $\mathcal{O}_t = \emptyset$, we recommend setting the GP prior mean of constraint $c$ to $\lambda_c$ in practice.

### 3.2 FUNCTION QUERY

To begin with, we provide insights into a rational function query selection strategy in the decoupled setting. Then, we will rigorously translate them into a concrete strategy.

---

[2]The term "vertical" refers to the output of $f$ often plotted as the (vertical) $y$-axis, as opposed to the term "horizontal" which refers to the input of $f$ often plotted as the (horizontal) $x$-axis, e.g., in Fig. 1a.

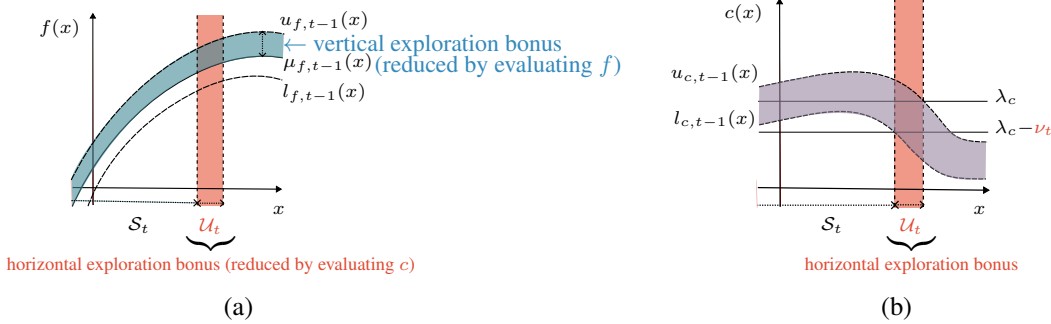

Figure 1: (a) Plot of the vertical and horizontal exploration bonuses in the space of the objective function; and (b) Plot of the horizontal exploration bonus in the space of the constraint function.

*Remark* 3.2 (On a rational function query selection strategy). In the decoupled setting, it can be inefficient by querying all functions in $\mathcal{F}$ at every iteration since the possibility of querying/evaluating functions in $\mathcal{F}$ independently is left unexploited. On one hand, it is unnecessary to evaluate any constraint at an input query that is likely to be feasible, which is illustrated in Fig. 2a in Sec. 4. On the other hand, if there is a significant risk of a constraint violation at $\mathbf{x}_t$, it is likely that querying the "most-violated" constraint eliminates $\mathbf{x}_t$ from the feasible region (hence, from being $\mathbf{x}^*$), in which case querying the objective function at $\mathbf{x}_t$ is redundant. It is illustrated in Figs. 2b-c in Sec. 4.

From the regret analysis perspective, the decoupled setting poses a challenge in bounding the instantaneous regret $r(\mathbf{x}_t)$ at each iteration. It is because $r(\mathbf{x}_t)$ depends on all instantaneous regrets $\{r_h(\mathbf{x}_t)\}_{h \in \mathcal{F}}$ while we do not evaluate all functions in $\mathcal{F}$ at each iteration, unlike in the coupled setting. Therefore, it is necessary to establish a connection across $\{r_h(\mathbf{x}_t)\}_{h \in \mathcal{F}}$.

To motivate our choice of $h_t$, we partition the optimistic feasible region $\mathcal{O}_t$ into a $\nu_t$-*relaxed feasible confidence region* $\mathcal{S}_t$ and an *uncharted region* $\mathcal{U}_t$.

$$\mathcal{O}_t = \mathcal{S}_t \cup \underbrace{\mathcal{U}_t}_{\text{horizontal exploration bonus}} \tag{12}$$

$$\mathcal{S}_t \triangleq \{\mathbf{x} \in \mathcal{X} | \, l_{c,t-1}(\mathbf{x}) \geq \lambda_c - \nu_t \, \forall c \in \mathcal{C}\} \cap \mathcal{O}_t \qquad \text{and} \qquad \mathcal{U}_t \triangleq \mathcal{O}_t \setminus \mathcal{S}_t \tag{13}$$

where $\nu_t \geq 0$ is a constraint-relaxation parameter ($\mathcal{S}_t$ and $\mathcal{U}_t$ are illustrated in Fig. 1b). Recall that $l_{c,t-1}(\mathbf{x}) \leq c(\mathbf{x})$ holds with high probability (Lemma 3.1), so does $\mathcal{S}_t \subset \widetilde{\mathcal{S}}_{\nu_t}$ where $\widetilde{\mathcal{S}}_{\nu_t} \triangleq \{\mathbf{x} \in \mathcal{X} | \, c(\mathbf{x}) \geq \lambda_c - \nu_t \, \forall c \in \mathcal{C}\}$ is a $\nu_t$-relaxation of $\mathcal{S}$. Therefore, $\mathcal{S}_t$ consists of feasible inputs w.r.t. $\widetilde{\mathcal{S}}_{\nu_t}$ with high probability. Furthermore, any input $\mathbf{x}$ in the uncharted region $\mathcal{U}_t$ satisfies

$$\exists c \in \mathcal{C}, \; u_{c,t-1}(\mathbf{x}) \geq \lambda_c \wedge \lambda_c - \nu_t > l_{c,t-1}(\mathbf{x}) \,. \tag{14}$$

Hence, the uncharted region $\mathcal{U}_t$ consists inputs whose feasibilities w.r.t. $\mathcal{S}$ are unknown (because $u_{c,t-1}(\mathbf{x}) \geq \lambda_c > l_{c,t-1}(\mathbf{x})$) and whose risks of a constraint violation are sufficiently high (because $\lambda_c - l_{c,t-1}(\mathbf{x}) \geq \nu_t$ for some $c \in \mathcal{C}$) as illustrated in Fig. 1b.

We interpret $\mathcal{U}_t$ as the *horizontal exploration bonus*. Its role in the optimistic feasible region $\mathcal{O}_t$ is analogous to the *vertical exploration bonus* $\beta_t^{1/2} \sigma_{f,t-1}(\mathbf{x})$ in the optimistic objective function evaluation $u_{f,t-1}(\mathbf{x})$ (illustrated in Fig. 1a and equation (8) vs. equation (12)). Let us translate the 2 intuitive cases in Remark 3.2 into the following concrete conditions.

**Querying a constraint when $\mathbf{x}_t \in \mathcal{U}_t$.** When $\mathbf{x}_t \in \mathcal{U}_t$, the risk of a constraint violation at $\mathbf{x}_t$ is sufficiently high ($\lambda_c - l_{c,t-1}(\mathbf{x}_t) \geq \nu_t$, illustrated in Fig. 1b). From Remark 3.2, we query the "most-violated" constraint defined as the one with the *highest risk of a constraint violation*, i.e., $\arg\max_{c \in \mathcal{C}} \lambda_c - l_{c,t-1}(\mathbf{x}_t)$. It is noted that as $\nu_t$ decreases, the size of the uncharted region $\mathcal{U}_t$ is non-decreasing (see Fig. 1b). Hence, we use $\nu_t$ to control the size of $\mathcal{U}_t$, i.e., controlling the horizontal exploration bonus. However, assigning *a small value* to $\nu_t$ is risky because we may excessively query a constraint, i.e., *excessive horizontal exploration*. Let us consider an extreme scenario: $\nu_t = 0$ and there exists an iteration $t$ such that a constraint $c \in \mathcal{C}$ is active at $\mathbf{x}_t$, i.e., $c(\mathbf{x}_t) = \lambda_c$. In this case, $\lambda_c - l_{c,t-1}(\mathbf{x}_t) = c(\mathbf{x}_t) - l_{c,t-1}(\mathbf{x}_t) \geq 0$, so the algorithm will keep querying a constraint without querying the objective function.

---

**Algorithm 1** UCB-D

---

**Require:** $\mathcal{X}, \mathcal{D}_0$
 1: Update GP posterior beliefs: $\{(\mu_{h,0}, \sigma_{h,0})\}_{h \in \mathcal{F}}$
 2: **for** $t \leftarrow 1; t \leftarrow t + 1; t \leq T$ **do**
 3:     $\mathbf{x}_t \leftarrow \arg\max_{\mathbf{x} \in \mathcal{O}_t} u_{f,t-1}(\mathbf{x})$
 4:     $c_t \leftarrow \arg\max_{c \in \mathcal{C}} \lambda_c - l_{c,t-1}(\mathbf{x}_t)$         `// most-violated constraint`
 5:     **if** $\lambda_{c_t} - l_{c_t,t-1}(\mathbf{x}_t) > 2\beta_t^{1/2}\sigma_{f,t-1}(\mathbf{x}_t)$ **then** `// ` $\mathbf{x}_t \in \mathcal{U}_t$
 6:         $h_t \leftarrow c_t$                             `// query most-violated constraint`
 7:     **else**                                        `// ` $\mathbf{x}_t \in \mathcal{S}_t$
 8:         $h_t \leftarrow f$                                `// query objective function`
 9:     **end if**
10:     $\mathbf{y}_{h_t}(\mathcal{D}_{h_t,t}) \leftarrow \mathbf{y}_{h_t}(\mathcal{D}_{h_t,t-1}) \cup \{y_{h_t}(\mathbf{x}_t)\}$
11:     Update GP posterior belief: $\mu_{h_t,t}, \sigma_{h_t,t}$
12: **end for**

---

**Querying the objective function when $\mathbf{x}_t \in \mathcal{S}_t$.** When $\mathbf{x}_t \in \mathcal{S}_t$, it is likely that $\mathbf{x}_t$ is a feasible solution (relaxed by $\nu_t$), illustrated in Fig. 1b. From Remark 3.2, we prefer querying the objective function. However, assigning *a large value* to $\nu_t$ is risky because we may excessively query the objective function, i.e., *insufficient horizontal exploration*. In particular, if $\nu_t > \max_{c \in \mathcal{C}} \max_{\mathbf{x} \in \mathcal{X}} \lambda_c - l_{c,t-1}(\mathbf{x})$, then $\mathcal{U}_t = \emptyset$ and $\mathcal{O}_t = \mathcal{S}_t$. It means $\mathbf{x}_t \in \mathcal{S}_t$ and the algorithm queries the objective function.

To resolve the dilemma of setting $\nu_t$ too large (excessively querying the objective function) or too small (excessively querying a constraint), we let $\nu_t$ to be "self-tuned" by tying its value with the vertical exploration bonus, i.e., setting $\nu_t = 2\beta_t^{1/2}\sigma_{f,t-1}(\mathbf{x}_t)$. The broad intuition is that if $\nu_t$ is too large, the algorithm repeats querying the objective function which reduces the vertical exploration bonus $\beta_t^{1/2}\sigma_{f,t-1}(\mathbf{x}_t)$. It, in turn, reduces $\nu_t$ as $\nu_t = 2\beta_t^{1/2}\sigma_{f,t-1}(\mathbf{x}_t)$. On the contrary, $\nu_t$ is too small only if $2\beta_t^{1/2}\sigma_{f,t-1}(\mathbf{x}_t)$ is too small. From equation (11), it implies that $r_f(\mathbf{x}_t)$ is small, so it is justifiable to refrain from querying the objective function. The resulting algorithm, called UCB-D, for the decoupled setting is described in Algorithm 1. In App. C, we prove the following Theorem 3.3 on the cumulative regret of Algorithm 1.

**Theorem 3.3.** *The cumulative regret $R_T$ of Algorithm 1 is bounded by*

$$Pr\left\{R_T \leq \sqrt{|\mathcal{F}|T\beta_T \max_{h \in \mathcal{F}} C_h \gamma_{h,T}} \; \forall T \geq 1\right\} \geq 1 - \delta \tag{15}$$

*where $C_h \triangleq 8/\log(1+\sigma_h^{-2})$ and $\gamma_{h,T}$ adopted from the work of Srinivas et al. (2010) is the maximum information gain from observing $T$ noisy evaluations of $h$.*

Srinivas et al. (2010) show that $\gamma_{h,T}$ is sublinear for some commonly used kernels including SE and Matérn kernels. Hence, Theorem 3.3 suggests that Algorithm 1 results in a cumulative regret that grows sublinearly when employing GPs with these kernels.

### 3.3 FUNCTION QUERY FROM ACTIVE LEARNING PERSPECTIVE

The choice of the input query $\mathbf{x}_t$ from GP-UCB exhibits an intriguing link to the concept of information gain found in the active learning literature, where one seeks the "most informative data point" or its approximate equivalent, the "most uncertain data point" as discussed in the work of Srinivas et al. (2010). Interestingly, one can view the choice of the function query $h_t$ as an uncertainty sampling strategy as well (i.e., seeking the "most uncertain data point") (Settles, 2009).

Let us denote the upper confidence bounds of the instantaneous regrets w.r.t the objective function $f$ and constraint $c$ in equation (11) as

$$u_{r_f,t-1}(\mathbf{x}_t) \triangleq 2\beta_t^{1/2}\sigma_{f,t-1}(\mathbf{x}_t) \geq r_f(\mathbf{x}_t) \tag{16}$$

$$u_{r_c,t-1}(\mathbf{x}_t) \triangleq \max(0, \lambda_c - l_{c,t-1}(\mathbf{x}_t)) \geq r_c(\mathbf{x}_t) . \tag{17}$$

While the nature of the uncertainty in active learning differs from that of the instantaneous regret in our problem, we are interested in minimizing their values in both scenarios. Hence, we employ the

uncertainty sampling paradigm to choose the function query by setting

$$h_t = \arg\max_{h \in \mathcal{F}} u_{r_h, t-1}(\mathbf{x}_t) . \tag{18}$$

This resulting strategy, interestingly, coincides with the choice of $h_t$ in Algorithm 1. It is noted that the uncertainty sampling approach may not be effective when there exists uncertainty that cannot be reduced with observations, referred to as *aleatoric uncertainty* in the work of Hüllermeier and Waegeman (2021). Fortunately, our instantaneous regrets are analogous to the *epistemic uncertainty* that can be reduced with observations. Specifically, the more observations we obtain, the better the objective function and constraints (hence, $\mathbf{x}^*$) are estimated. Thus, a smaller regret can be achieved.

*Remark* 3.4. If the evaluation of $h \in \mathcal{F}$ incurs a cost $l(h) > 0$, then we would like to choose the function query by maximizing a *cost-aware* upper confidence bound of the instantaneous regret. Specifically, the function query is chosen as $h_t = \arg\max_{h \in \mathcal{F}} u_{r_h, t-1}(\mathbf{x}_t)/l(h)$. It is interpreted as the upper confidence bound of the instantaneous regret per unit cost (Swersky et al., 2013).

## 3.4 ESTIMATOR OF THE OPTIMAL SOLUTION

Remark 2.1 states that $s(\mathbf{x})$ is more effective than $r(\mathbf{x})$ in assessing the suboptimality of a solution. Hence, we would like to use $s(\mathbf{x})$ to propose an estimator $\tilde{\mathbf{x}}_t^*$ for approximating the optimal solution $\mathbf{x}^*$. From equation (11), we obtain an upper confidence bound of $s(\mathbf{x})$ at iteration $t$

$$s(\mathbf{x}) \triangleq \sum_{h \in \mathcal{F}} r_h(\mathbf{x}) \leq \sum_{h \in \mathcal{F}} u_{r_h, t-1}(\mathbf{x}) \tag{19}$$

where $u_{r_h, t-1}$ is defined in equation (16) and equation (17). We would like to select the input with the lowest upper confidence bound of $s(\mathbf{x})$ as the estimator by considering all previous $t-1$ iterations

$$\tilde{\mathbf{x}}_t^* = \tilde{\mathbf{x}}_{\kappa(t)} , \tag{20}$$

where

$$\tilde{\mathbf{x}}_{t'} \triangleq \arg\min_{\mathbf{x} \in \mathcal{X}} \sum_{h \in \mathcal{F}} u_{r_h, t'-1}(\mathbf{x}) \quad \text{and} \quad \kappa(t) \triangleq \arg\min_{t'=1,\ldots,t} \sum_{h \in \mathcal{F}} u_{r_h, t'-1}(\tilde{\mathbf{x}}_{t'}) . \tag{21}$$

Then, App. D proves the following lemma.

**Lemma 3.5.** *By picking the estimator in equation (20), it holds with probability $\geq 1 - \delta$ that*

$$\forall t \geq 1, \ s(\tilde{\mathbf{x}}_t^*) \leq |\mathcal{F}| \sqrt{|\mathcal{F}| \beta_t \max_{h \in \mathcal{F}} C_h \gamma_{h,t}/t} \tag{22}$$

*where $C_h \triangleq 8/\log(1 + \sigma_h^{-2})$ and $\gamma_{h,T}$ adopted from the work of Srinivas et al. (2010) is the maximum information gain from observing $T$ noisy evaluations of $h$.*

Hence, when $\gamma_{h,t}$ is sublinear (e.g., when the kernel is SE or Matérn (Srinivas et al., 2010)), the sum $s(\tilde{x}_t^*)$ of instantaneous regrets at the estimator approaches 0 as $t \to \infty$.

## 4 EXPERIMENTS

This section validates the empirical performance of our algorithms (UCB-C in the coupled setting and UCB-D in the decoupled setting) by comparing with EIC (Gardner et al., 2014), ADMMBO (Ariafar et al., 2019), and the state-of-the-art CMES-IBO which significantly outperforms other existing approaches including EIC and PESC in the work of Takeno et al. (2022). We did not conduct a comparison with PESC due to the challenge of maintaining a consistent initial configuration for PESC, as emphasized by Takeno et al. (2022). Moreover, in the decoupled setting, PESC is not currently available in the primary branch of the Spearmint tool at `https://github.com/HIPS/Spearmint`. This also highlights the complexity involved in implementing PESC for decoupled queries and its limited accessibility to practitioners. For these baselines, we select the estimator $\tilde{\mathbf{x}}_t^* = \arg\max_{\mathbf{x} \in \mathcal{X}} \mu_{f,t-1}(\mathbf{x})$ such that $\forall c \in \mathcal{C}, \Pr(c(\mathbf{x}) \geq \lambda_c) \geq \sqrt[|\mathcal{C}|]{0.95}$ as suggested by Takeno et al. (2022). This definition may be undefined in the presence of an equality constraint, so we do not consider equality constraints in our experiments. The estimator in our algorithms is described in equation (20). To illustrate both the instantaneous regrets of the objective function and constraints, we plot the average and standard error (over 10 repeated experiments) of the sum $s(\tilde{\mathbf{x}}_t^*)$ (equation (5)) of these regrets at the estimator against the number of queries $|\mathcal{D}_t|$. The noise's standard deviation is set at $\sigma_h = 0.01 \ \forall h \in \mathcal{F}$. Additional details are described in App. E.

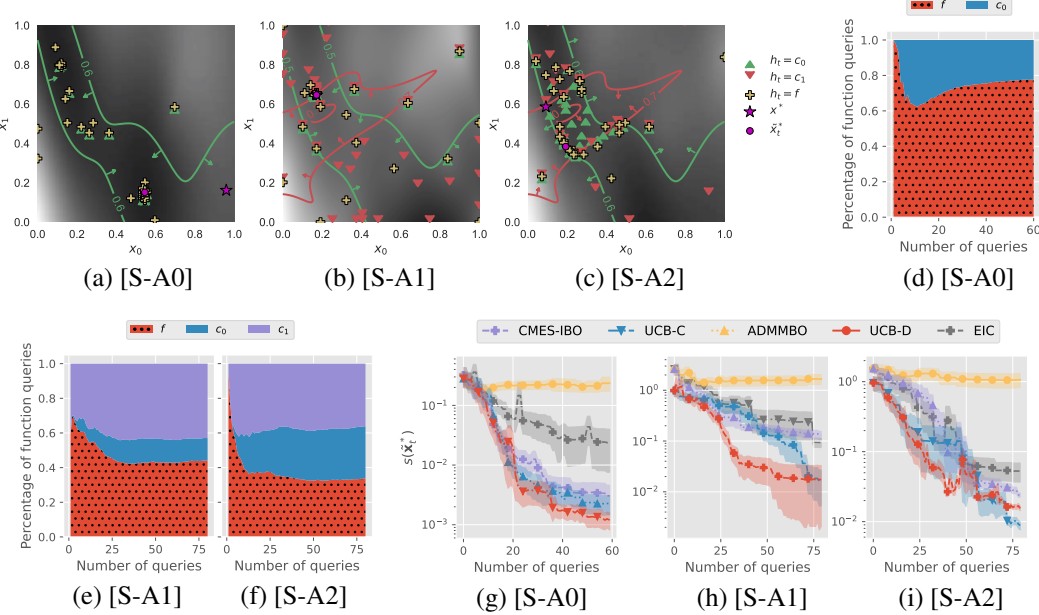

Figure 2: Synthetic problems: (a-c) Plots of the upper confidence bound $u_{f,t-1}$ (plotted as the gray heatmap), constraints (plotted as contour lines with arrows showing the side of the feasible region), the optimal $x^*$, the estimator $\tilde{\mathbf{x}}_t^*$, and input queries of UCB-D; (d-f) Plots of the percentage of the objective function (the dotted area) and constraints selected as $h_t$ by UCB-D; (g-i) Plots of the instantaneous regret $s(\tilde{\mathbf{x}}_t^*)$ against the number of queries.

## 4.1 SYNTHETIC PROBLEMS

The experiments are conducted on 3 synthetic constrained optimization problems each labeled in the format [S-A{#_of_active_constraints_at_$\mathbf{x}^*$}]: [S-A0], [S-A1], and [S-A2] with 0, 1, and 2 active constraints, respectively (the formulations are described in App. E.1). In these problems, there are 2 input dimensions so we can visualize the constraints and the input queries as shown in Figs. 2a-c. **[S-A0]:** The constraint is inactive at $\mathbf{x}^*$ which is located distant from the boundary of the feasible region $\mathcal{S}$ (Fig. 2a). Thus, UCB-D does not require precise boundary estimations of $\mathcal{S}$ (i.e., of $c_0$) to pinpoint $\mathbf{x}^*$. This results in a sparse allocation of input queries around the boundary of $\mathcal{S}$. Furthermore, around the estimator $\tilde{\mathbf{x}}_t^*$, a substantial number of queries are evaluated at the objective function (i.e., $h_t = f$, as plotted by the yellow pluses) due to the high certainty that the input query is feasible, which aligns with Remark 3.2. Specifically, Fig. 2d shows that more than $70\%$ of 60 input queries are evaluated at the objective function $f$, as plotted by the dotted area. Despite the distance between the estimator $\tilde{\mathbf{x}}_t^*$ and the optimal $\mathbf{x}^*$, the difference between $f(\tilde{\mathbf{x}}_t^*)$ and $f(\mathbf{x}^*)$ is minimal because $s(\tilde{\mathbf{x}}_t^*)$ is small in Fig. 2g for UCB-D. **[S-A1]:** At $\mathbf{x}^*$, the constraint $c_0$ is inactive, but unlike [S-A0], the constraint $c_1$ is active (Fig. 2b). Thus, it requires precise boundary estimations of $c_1$ around $\mathbf{x}^*$ to pinpoint $\mathbf{x}^*$, but does not require precise boundary estimations of $c_0$. This results in a sparse allocation of input queries around the boundary of $c_0$ and a denser allocation of input queries around the boundary of $c_1$, especially near $\tilde{\mathbf{x}}_t^*$ in Fig. 2b. Specifically, Fig. 2e shows that only a small number of the input queries are evaluated at the inactive $c_0$. **[S-A2]:** Both constraints $c_0$ and $c_1$ are active at $\mathbf{x}^*$ (Fig. 2c). Thus, UCB-D requires precise boundary estimations of both $c_0$ and $c_1$ around $\mathbf{x}^*$ to pinpoint $\mathbf{x}^*$. This results in a dense allocation of input queries around the boundaries of both $c_0$ and $c_1$, especially around $\mathbf{x}^*$ in Fig. 2c. Specifically, Fig. 2f shows that the input queries are roughly allocated equally to the objective function and the 2 constraints. Despite the distance between $\tilde{\mathbf{x}}_t^*$ and $\mathbf{x}^*$, the difference between $f(\tilde{\mathbf{x}}_t^*)$ and $f(\mathbf{x}^*)$ is minimal because $s(\tilde{\mathbf{x}}_t^*)$ is small in Fig. 2i for UCB-D. In Figs. 2b-c, we also observe that only a minority of the function queries $h_t = f$ are located far away from the feasible region $\mathcal{S}$, which aligns with Remark 3.2.

Regarding the instantaneous regret, Figs. 2g-i show that our UCB-D converges faster than other algorithms. Hence, UCB-D is more query-efficient, as explained by the above discussion. UCB-C performs competitively compared to the state-of-the-art CMES-IBO as both are designed for the

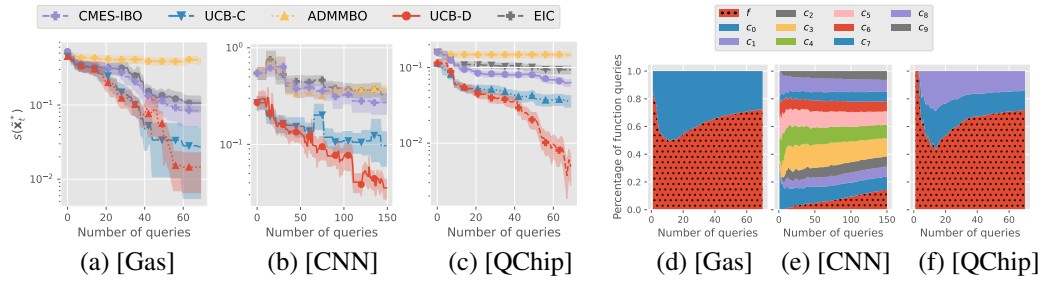

Figure 3: Real-world problems: Plots of (a-c) the instantaneous regret $s(\tilde{\mathbf{x}}^*)$ and (d-f) the percentage of function queries (by UCB-D) against the number of queries.

coupled setting. ADMMBO does not work well probably because it requires tuning the number of evaluations of $f$ and $c$ at each BO iteration. In our experiments, ADMMBO evaluates $f$ and $c$ once at each BO iteration to be consistent with EIC, CMES-IBO, and UCB-C.

## 4.2 REAL-WORLD PROBLEMS

In this section, we introduce 3 optimization problems utilizing real-world objective functions and constraints. These problems serve to assess the effectiveness of our algorithms in practice. We select a real-world problem of optimizing a gas transmission compressor design, referred to as [Gas], from Kumar et al. (2020). It consists of $d = 4$ input dimensions and has $|\mathcal{C}| = 1$ constraint. The problem of tuning hyperparameters of a convolutional neural network (CNN), referred to as [CNN], is taken from the work of Takeno et al. (2022). In the [CNN] problem, a two-layer CNN is trained on a class-imbalanced CIFAR10 dataset. The goal is to maximize the overall accuracy across 10 classes subject to the constraint that the recall of each class is at least 0.5, i.e., $|\mathcal{C}| = 10$. There are $d = 5$ hyperparameters to be optimized. The final experiment, referred to as [QChip], involves maximizing the coupling strength of a synthetic superconducting quantum chip (Yan et al., 2018). While it is a critical aspect for the chip's performance, coupling strength must be carefully controlled within the constraints of the coupling energy to prevent issues like noise, cross-talk between qubits, and poor gate fidelity (Kwon et al., 2021). In particular, we maximize the coupling strength subject to $|\mathcal{C}| = 2$ constraints specifying the desirable range of the coupling energy, by adjusting $d = 11$ geometric features that describe the physical dimensions and arrangement of quantum chip components. To create the groundtruth functions, we obtain a dataset consisting of 393 data points. They are rigorously generated through extensive simulations using electrical simulation software (see App. E.3).

Figs. 3(a-c) show that our UCB-D and UCB-C converge faster than other baseline methods. Therefore, when considering the same number of queries, our algorithms outperform other baseline methods in identifying superior designs for the above real-world problems. In Figs. 3d and 3f, we observe that the number of queries to the objective function dominates that to the constraints in the [Gas] and [QChip] experiments, hinting that the constraints are inactive at the optimal solution, aligning with the groundtruth. Fig. 3e shows that in the [CNN] experiment, UCB-D initially focuses on identifying a feasible input as it allocates few queries to the objective function at the start. It is noted that locating a feasible input is more challenging in this experiment due to the large number of constraints.

## 5 CONCLUSION

In this paper, we propose a novel constrained BO algorithm with a provable performance guarantee that adaptively selects not only the input query but also the function query to account for the decoupled query. We formulate the algorithm from the standpoint of the fundamental exploration-exploitation trade-off and, interestingly, cast the proposed algorithm under the uncertainty sampling paradigm in the active learning literature. As our constrained BO solution requires only the confidence bounds of the function evaluations, we believe the approach can be applied to other BO problems such as BO of risk measures (Cakmak et al., 2020; Nguyen et al., 2021b;a) and meta-BO (Nguyen et al., 2023).

## REPRODUCIBILITY STATEMENT

We have described detailed proofs for the theoretical results in App. B, C, and D. These proofs utilize an assumption from the work of Srinivas et al. (2010) as elaborated in Sec. 3. Regarding the experimental results, we have included both the code and the datasets in the submission. We have also provided a more detailed description of the experiment settings in App. E.

## ACKNOWLEDGEMENT

This research is supported by AI Singapore, under grant AISG2-RP-2020-018.

This research is part of the programme DesCartes and is supported by the National Research Foundation, Prime Minister's Office, Singapore under its Campus for Research Excellence and Technological Enterprise (CREATE) programme.

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
