## A ON ISSUE WITH EQUALITY CONSTRAINT

Several existing works including EIC (Gardner et al., 2014) and CMES-IBO (Takeno et al., 2022) rely on the probability mass $P(c(\mathbf{x}) \geq \lambda_c \ \forall c \in \mathcal{C})$ where the unknown constraint $c$ is modeled with a GP. Let us consider an optimization problem where there is only 1 equality constraint $c(\mathbf{x}) = 0$. Then, the above probability mass $P(c(\mathbf{x}) = 0)$ is 0. As a result, existing works such as CMES-IBO require manually adding a tolerance value to ensure the probability mass is strictly positive.

Regarding PESC (Hernández-Lobato et al., 2016), it relies on the sampling of $\mathbf{x}^*$. When there are several equality constraints, the probability of obtaining a feasible solution is small, making the sampling inefficient.

In our approach, we consider an optimistic feasible region (which is partitioned into $\mathcal{S}_t$ and $\mathcal{U}_t$ shown in Fig. 1b) that can handle equality constraints without any modification. Though one may argue that the role of $\nu_t$ is similar to the tolerance added to the equality constraint in CMES-IBO, it is noted that $\nu_t$ is not fixed to any pre-defined sequence but it is "self-tuned" by the vertical exploration bonus as explained in Sec. 3.2 and has a theoretical justification.

When proposing an estimator for approximating the optimal solution, one may consider a pessimistic feasible region to ensure the feasbility of the estimator with high probability. However, the pessimistic feasible region can be empty if the optimal solution is not an interior point of the feasible region, e.g., when there are equality constraints.

## B A BAYESIAN OPTIMIZATION ALGORITHM FOR COUPLED QUERIES

---
**Algorithm 2** UCB-C
---
**Require:** $\mathcal{X}, \mathcal{D}_0$
1: Update GP posterior beliefs: $\{(\mu_{h,0}, \sigma_{h,0})\}_{h \in \mathcal{F}}$
2: **for** $t \leftarrow 1; t \leftarrow t + 1; t \leq T$ **do**
3:     $\mathbf{x}_t \leftarrow \arg\max_{\mathbf{x} \in \mathcal{O}_t} u_{f,t-1}(\mathbf{x})$
4:     **for** $h \in \mathcal{F}$ **do**               // coupled query
5:         $\mathbf{y}_h(\mathcal{D}_{h,t}) \leftarrow \mathbf{y}_h(\mathcal{D}_{h,t-1}) \cup \{y_h(\mathbf{x}_t)\}$
6:         Update GP posterior belief: $\mu_{h,t}, \sigma_{h,t}$
7:     **end for**
8: **end for**
---

Our proposed algorithm for coupled queries is shown in Algorithm 2. In this section, we assume that $h(\mathbf{x}) \in [l_{h,t-1}(\mathbf{x}), u_{h,t-1}(\mathbf{x})]$ for all $h \in \mathcal{F}$, $\mathbf{x} \in \mathcal{X}$, and $t \geq 1$, which happens with probability $\geq 1 - \delta$ from Lemma 3.1. In order to prove the upper confidence bound of its cumulative regret, we derive the inequalities in equation (11) as follows.

The instantaneous regret w.r.t. the objective function $f$ is bounded by

$$r_f(\mathbf{x}_t) \triangleq \max(0, f(\mathbf{x}^*) - f(\mathbf{x}_t)) \tag{23}$$
$$\leq \max(0, u_{f,t-1}(\mathbf{x}^*) - l_{f,t-1}(\mathbf{x}_t)) \quad \text{from Lemma 3.1} \tag{24}$$
$$\leq \max(0, u_{f,t-1}(\mathbf{x}_t) - l_{f,t-1}(\mathbf{x}_t)) \tag{25}$$
$$= u_{f,t-1}(\mathbf{x}_t) - l_{f,t-1}(\mathbf{x}_t) \tag{26}$$
$$= 2\beta_t^{1/2} \sigma_{f,t-1}(\mathbf{x}_t) \tag{27}$$

where inequality equation (25) holds with probability $\geq 1 - \delta$ as

- $\mathbf{x}^* \in \mathcal{O}_t$ with probability $\geq 1 - \delta$ since $\mathcal{O}_t \supset \mathcal{S}$ with probability $\geq 1 - \delta$.

- $u_{f,t-1}(\mathbf{x}_t) \geq u_{f,t-1}(\mathbf{x})$ for all $\mathbf{x} \in \mathcal{O}_t$ because $\mathbf{x}_t \triangleq \arg\max_{\mathbf{x} \in \mathcal{O}_t} u_{f,t-1}(\mathbf{x})$.

The instantaneous regret w.r.t. the constraint $c$ is bounded by

$$r_c(\mathbf{x}_t) \triangleq \max(0, \lambda_c - c(\mathbf{x}_t)) \tag{28}$$
$$\leq \max(0, \lambda_c - l_{c,t-1}(\mathbf{x}_t)) \quad \text{from Lemma 3.1} \tag{29}$$
$$\leq \max(0, u_{c,t-1}(\mathbf{x}_t) - l_{c,t-1}(\mathbf{x}_t)) \tag{30}$$
$$= u_{c,t-1}(\mathbf{x}_t) - l_{c,t-1}(\mathbf{x}_t) \tag{31}$$
$$= 2\beta_t^{1/2}\sigma_{c,t-1}(\mathbf{x}_t) \tag{32}$$

where inequality equation (30) holds with probability $\geq 1 - \delta$ as $u_{c,t-1}(\mathbf{x}_t) \geq \lambda_c$ because $\mathbf{x}_t \in \mathcal{O}_t$. Therefore,

$$r(\mathbf{x}_t) \triangleq \max_{h \in \mathcal{F}} r_h(\mathbf{x}_t) \leq \max_{h \in \mathcal{F}} 2\beta_t^{1/2}\sigma_{h,t-1}(\mathbf{x}_t) \tag{33}$$

$$R_T \triangleq \sum_{t=1}^{T} r(\mathbf{x}_t) \leq \sum_{t=1}^{T} \max_{h \in \mathcal{F}} 2\beta_t^{1/2}\sigma_{h,t-1}(\mathbf{x}_t) \,. \tag{34}$$

Let $T_h \triangleq \sum_{t=1}^{T} \mathbb{1}_{h = \arg\max_{h' \in \mathcal{F}} \sigma_{h',t-1}(\mathbf{x}_t)}$ (breaking the tie arbitrarily if necessary to ensure a unique maximizer) which implies that $\sum_{h \in \mathcal{F}} T_h = T$.

$$R_T \leq \sum_{t=1}^{T} \max_{h \in \mathcal{F}} 2\beta_t^{1/2}\sigma_{h,t-1}(\mathbf{x}_t) \tag{35}$$

$$\leq \sum_{h \in \mathcal{F}} 2\beta_T^{1/2} \sum_{t=1}^{T_h} \sigma_{h,t-1}(\mathbf{x}_t) \tag{36}$$

as $\beta_t$ is non-decreasing. Furthermore, from Lemma 5.4 in Srinivas et al. (2010):

$$\sum_{t=1}^{T_h} \sigma_{h,t-1}^2(\mathbf{x}_t) \leq C_h \gamma_{h,T_h}/4 \tag{37}$$

where $C_h \triangleq 8/\log(1 + \sigma_h^{-2})$ and $\gamma_{h,T_h}$ is the maximum information gain from observing $T_h$ noisy evaluations of $h$. Therefore, applying the Cauchy-Schwarz inequality,

$$\sum_{t=1}^{T_h} \sigma_{h,t-1}(\mathbf{x}_t) \leq \sqrt{T_h \sum_{t=1}^{T_h} \sigma_{h,t-1}^2(\mathbf{x}_t)} \leq \sqrt{T_h C_h \gamma_{h,T_h}/4} \,. \tag{38}$$

Hence,

$$R_T \leq \sum_{h \in \mathcal{F}} 2\beta_T^{1/2}\sqrt{T_h C_h \gamma_{h,T_h}/4}$$
$$\leq \beta_T^{1/2} \sum_{h \in \mathcal{F}} \sqrt{T_h}\sqrt{C_h \gamma_{h,T_h}}$$
$$\leq \beta_T^{1/2}\sqrt{\left(\sum_{h \in \mathcal{F}} T_h\right)\left(\sum_{h' \in \mathcal{F}} C_{h'}\gamma_{h',T_{h'}}\right)} \quad \text{Cauchy-Schwarz inequality}$$
$$= \sqrt{T\beta_T \sum_{h' \in \mathcal{F}} C_{h'}\gamma_{h',T_{h'}}}$$
$$\leq \sqrt{T\beta_T|\mathcal{F}| \max_{h \in \mathcal{F}} C_h \gamma_{h,T_h}}$$
$$\leq \sqrt{|\mathcal{F}|T\beta_T \max_{h \in \mathcal{F}} C_h \gamma_{h,T}} \,.$$

## C    PROOF OF THEOREM 3.3

From Algorithm 1, we observe that:

**Case 1.** When $h_t = f$, we show that the instantaneous regrets w.r.t. the objective function and constraints are bounded by $2\beta_t^{1/2}\sigma_{f,t-1}(\mathbf{x}_t)$ as follows.

$$r_f(\mathbf{x}_t) \leq 2\beta_t^{1/2}\sigma_{f,t-1}(\mathbf{x}_t) \quad \text{from equation (11) .}$$

Recall that we select $h_t = f$ when $\mathbf{x}_t \in \mathcal{S}_t$, i.e., for all $c \in \mathcal{C}$, $\lambda_c - l_{c,t-1}(\mathbf{x}_t) \leq \nu_t = 2\beta_t^{1/2}\sigma_{f,t-1}(\mathbf{x}_t)$, so

$$
\begin{aligned}
\forall c \in \mathcal{C}, r_c(\mathbf{x}_t) &\triangleq \max(0, \lambda_c - c(\mathbf{x}_t)) \\
&\leq \max(0, \lambda_c - l_{c,t-1}(\mathbf{x}_t)) \quad \text{from Lemma 3.1} \\
&\leq \max(0, 2\beta_t^{1/2}\sigma_{f,t-1}(\mathbf{x}_t)) \\
&= 2\beta_t^{1/2}\sigma_{f,t-1}(\mathbf{x}_t) .
\end{aligned}
$$

Therefore, when $h_t = f$,

$$r(\mathbf{x}_t) \triangleq \max_{h \in \mathcal{F}} r_h(\mathbf{x}_t) \leq 2\beta_t^{1/2}\sigma_{f,t-1}(\mathbf{x}_t) . \tag{39}$$

**Case 2.** When $h_t = c$, we show that the instantaneous regrets w.r.t. the objective function and constraints are bounded by $2\beta_t^{1/2}\sigma_{c,t-1}(\mathbf{x}_t)$ as follows.

$$\forall c \in \mathcal{C}, r_c(\mathbf{x}_t) \leq 2\beta_t^{1/2}\sigma_{c,t-1}(\mathbf{x}_t) \quad \text{from equation (11) .}$$

Recall that we select $h_t = c$ when $\mathbf{x}_t \in \mathcal{U}_t$, i.e., $\exists c \in \mathcal{C}$, $\lambda_c - l_{c,t-1}(\mathbf{x}_t) > \nu_t = 2\beta_t^{1/2}\sigma_{f,t-1}(\mathbf{x}_t)$. This implies that

$$
\begin{aligned}
2\beta_t^{1/2}\sigma_{f,t-1}(\mathbf{x}_t) &< \max_{c \in \mathcal{C}} \lambda_c - l_{c,t-1}(\mathbf{x}_t) \\
&= \lambda_{c_t} - l_{c_t,t-1}(\mathbf{x}_t) \quad \text{where } c_t \text{ is defined in Algorithm 1 .}
\end{aligned}
$$

$$
\begin{aligned}
r_f(\mathbf{x}_t) &\leq 2\beta_t^{1/2}\sigma_{f,t-1}(\mathbf{x}_t) \quad \text{from equation (11)} \\
&< \lambda_{c_t} - l_{c_t,t-1}(\mathbf{x}_t) \\
&\leq u_{c_t,t-1}(\mathbf{x}_t) - l_{c_t,t-1}(\mathbf{x}_t) \quad \text{as } \mathbf{x}_t \in \mathcal{O}_t, \text{ i.e., } u_{c,t-1}(\mathbf{x}_t) \geq \lambda_c \; \forall c \in \mathcal{C} \\
&= 2\beta_t^{1/2}\sigma_{c_t,t-1}(\mathbf{x}_t) .
\end{aligned}
$$

Therefore, when $h_t = c$,

$$r(\mathbf{x}_t) \triangleq \max_{h \in \mathcal{F}} r_h(\mathbf{x}_t) \leq 2\beta_t^{1/2}\sigma_{c_t,t-1}(\mathbf{x}_t) . \tag{40}$$

Combining the above 2 cases in equation (39) and equation (40),

$$r(\mathbf{x}_t) \leq 2\beta_t^{1/2}\sigma_{h_t,t-1}(\mathbf{x}_t) . \tag{41}$$

Hence,

$$R_T \triangleq \sum_{t=1}^{T} r(\mathbf{x}_t) \leq \sum_{t=1}^{T} 2\beta_t^{1/2}\sigma_{h_t,t-1}(\mathbf{x}_t) . \tag{42}$$

Let $T_h' \triangleq \sum_{t=1}^{T} \mathbb{1}_{h=h_t}$, then using the non-decreasing property of $\beta_t$, we can rewrite the above inequality as

$$R_T \leq \sum_{h \in \mathcal{F}} 2\beta_T^{1/2} \sum_{t=1}^{T_h'} \sigma_{h,t-1}(\mathbf{x}_t) . \tag{43}$$

The above result is the same as equation (36). Therefore, we can follow the argument in App. B to obtain the same upper confidence bound of $R_T$:

$$R_T \leq \sqrt{|\mathcal{F}|T\beta_T \max_{h \in \mathcal{F}} C_h \gamma_{h,T}} . \tag{44}$$

# D   PROOF OF LEMMA 3.5

App. B and C both show that with probability $\geq 1 - \delta$,

$$\forall t \geq 1, \ \sum_{t'=1}^{t} \max_{h \in \mathcal{F}} u_{r_h, t'-1}(\mathbf{x}_{t'}) \leq \sqrt{|\mathcal{F}| t \beta_t \max_{h \in \mathcal{F}} C_h \gamma_{h,t}} \tag{45}$$

where $u_{r_h, t-1}(\mathbf{x})$ is defined in equation (16) and equation (17). Equivalently,

$$\sqrt{|\mathcal{F}| \beta_t \max_{h \in \mathcal{F}} C_h \gamma_{h,t} / t} \geq \frac{1}{t} \sum_{t'=1}^{t} \max_{h \in \mathcal{F}} u_{r_h, t'-1}(\mathbf{x}_{t'}) \tag{46}$$

$$\geq \frac{1}{t} \sum_{t'=1}^{t} \frac{1}{|\mathcal{F}|} \sum_{h \in \mathcal{F}} u_{r_h, t'-1}(\mathbf{x}_{t'}) \tag{47}$$

$$\geq \frac{1}{t} \sum_{t'=1}^{t} \min_{\mathbf{x} \in \mathcal{X}} \frac{1}{|\mathcal{F}|} \sum_{h \in \mathcal{F}} u_{r_h, t'-1}(\mathbf{x}) \tag{48}$$

$$\geq \min_{t'=1,\ldots,t} \min_{\mathbf{x} \in \mathcal{X}} \frac{1}{|\mathcal{F}|} \sum_{h \in \mathcal{F}} u_{r_h, t'-1}(\mathbf{x}) \ . \tag{49}$$

Let

$$\tilde{\mathbf{x}}_{t'} \triangleq \arg \min_{\mathbf{x} \in \mathcal{X}} \sum_{h \in \mathcal{F}} u_{r_h, t'-1}(\mathbf{x}) \tag{50}$$

$$\kappa(t) \triangleq \arg \min_{t'=1,\ldots,t} \sum_{h \in \mathcal{F}} u_{r_h, t'-1}(\tilde{\mathbf{x}}_t) \ . \tag{51}$$

Our estimator is chosen as

$$\tilde{\mathbf{x}}_t^* = \tilde{\mathbf{x}}_{\kappa(t)} \ , \tag{52}$$

then

$$\min_{t'=1,\ldots,t} \min_{\mathbf{x} \in \mathcal{X}} \frac{1}{|\mathcal{F}|} \sum_{h \in \mathcal{F}} u_{r_h, t'-1}(\mathbf{x}) = \frac{1}{|\mathcal{F}|} \sum_{h \in \mathcal{F}} u_{r_h, \kappa(t)-1}(\tilde{\mathbf{x}}_t^*) \geq \frac{1}{|\mathcal{F}|} \sum_{h \in \mathcal{F}} r_h(\tilde{\mathbf{x}}_t^*) = \frac{s(\tilde{\mathbf{x}}_t^*)}{|\mathcal{F}|} \tag{53}$$

where the inequality holds with probability $\geq 1 - \delta$. Hence,

$$\Pr\left\{ s(\tilde{\mathbf{x}}_t^*) \leq |\mathcal{F}| \sqrt{|\mathcal{F}| \beta_t \max_{h \in \mathcal{F}} C_h \gamma_{h,t} / t} \right\} \geq 1 - \delta \ . \tag{54}$$

# E   ADDITIONAL EXPERIMENT DETAILS

We refrain from conducting a comparison with EPBO and PESC for the following reasons. EPBO is equivalent to our UCB-C method when an appropriate value of $\rho$ is chosen. However, an automated strategy of selecting $\rho$ is left unspecified in the work of Lu and Paulson (2022). On the other hand, the noteworthy advantage of our proposed UCB-C approach is its ability to overcome the need of selecting such a parameter. Regarding PESC in the coupled setting, Takeno et al. (2022) raises the difficulty in implementing and assigning the initial configuration in the PESC package (Spearmint). This difficulty poses obstacles to achieving consistent initial experiment configurations for PESC. Regarding PESC in the decoupled setting, it is not included in the main branch of Spearmint. Therefore, we opt not to use PESC as a baseline in the experiment. On the other hand, the work of Takeno et al. (2022) shows that CMES-IBO outperforms PESC by a large margin in the coupled setting. Hence, we demonstrate the performance of our algorithms by comparing with CMES-IBO.

To ensure that the global optimal solution can be identified and to prevent variations in performance among different methods stemming from being trapped in distinct local optima when using continuous optimization tools, we discretize the input space into 10,000 randomly selected input points in the experiments: [S-A0], [S-A1], [S-A2], [Gas], and [Beam]. For the [CNN] and [QChip] experiments,

the size of the input space matches that of the generated dataset, i.e., 5120 and 393, respectively. In practice, our proposed methods can be implemented in a continuous input domain using any continuous constrained optimization package because only step 3 (selecting $\mathbf{x}_t$) in both Algorithms 1 and 2 involves solving a constrained optimization problem (with a known objective function and constraint).

The GP hyperparameters including the parameters of the SE kernel and the noise variance $\{\sigma_h^2\}_{h \in \mathcal{F}}$ are assumed to be unknown in our experiments. We optimize them after every BO iteration by maximizing the likelihood of the observations using Adam optimizer. We assign prior distributions to the GP hyperparameters to avoid numerical issues when performing the likelihood maximization. The prior distributions of the length-scale, the signal standard deviation, and the noise standard deviation are $\mathrm{Gamma}(0.25, 0.5)$, $\mathrm{Gamma}(2, 0.15)$, and $\mathcal{N}(0.0, 0.1)$, respectively. Furthermore, the noise standard deviation is constrained to be at least $0.01$.

To ensure the initial observations are consistent with the coupled setting, we initialize all experiments with coupled observations, i.e., the evaluations of the objective function and constraints are at the same set of inputs: $\mathcal{D}_{f,0} = \mathcal{D}_{c,0}$ for all $c \in \mathcal{C}$. In the synthetic experiments, the number of initial coupled observations are 3, 5, and 5 for [S-A0], [S-A1], and [S-A2], respectively. In the real-world experiments, the number of initial coupled observations are 7, 30, and 7 for [Gas], [CNN], and [QChip], respectively.

Throughout the remainder of this section, we provide more detailed descriptions of several experiments.

### E.1 SYNTHETIC EXPERIMENTS

Let $g_b$ and $g_g$ denote the Branin-Hoo and the Goldstein-Price functions where the input domain is normalized to range $[0, 1]^2$. They are obtained from `https://www.sfu.ca/~ssurjano`.

Then, the [S-A0] problem is defined as

$$\max_{\mathbf{x}} \; g_b(\mathbf{x}) \text{ s.t. } g_b(\mathbf{x}) \geq 0.6 \;.$$

The [S-A1] problem is defined as

$$\max_{\mathbf{x}} g_b(\mathbf{x})$$
$$\text{s.t. } g_b(\mathbf{x}) \geq 0.5$$
$$g_g(\mathbf{x}) \geq 0.7 \;.$$

The [S-A2] problem is defined as

$$\max_{\mathbf{x}} g_b(\mathbf{x})$$
$$\text{s.t. } g_b(\mathbf{x}) \leq 0.6$$
$$g_g(\mathbf{x}) \geq 0.7 \;.$$

It is noted that although the same function is used in both the objective function and a constraint in the above optimization problems, we treat them as distinct black-box functions and model them with independent GPs.

### E.2 GAS TRANSMISSION COMPRESSOR DESIGN (KUMAR ET AL., 2020)

There are $d = 4$ input dimension: $\mathbf{x} = (x_i)_{i=1}^4$ with the following bounds:

$$20 \leq x_1 \leq 50$$
$$1 \leq x_2 \leq 10$$
$$20 \leq x_3 \leq 50$$
$$0.1 \leq x_4 \leq 60 \;.$$

The objective function and the constraint are specified as follows.

$$f(\mathbf{x}) = 8.16 \times 10^5 x_1^{1/2} x_2 x_3^{-2/3} x_4^{-1/2} + 3.69 \times 10^4 x_3 + 7.72 \times 10^8 x_1^{-1} x_2^{0.219} - 765.43 \times 10^6 x_1^{-1}$$

$$c(\mathbf{x}) = x_4 x_2^{-2} + x_2^{-2} - 1 \leq 0 .$$

We normalize the range of the objective function and the constraint to the range $[-1, 1]$ and the input to to domain $[0, 1]^4$.

### E.3 MAXIMIZING COUPLING STRENGTH OF SYNTHETIC SUPERCONDUCTING QUANTUM CHIP

We address the optimization problem of fine-tuning the critical parameter of coupling strength in a synthetic superconducting quantum chip (Yan et al., 2018). This optimization problem is motivated by the fundamental role that coupling strength plays in influencing the chip's performance (Sete et al., 2021; Wu et al., 2021). Specifically, the coupling strength affects the speed and fidelity of quantum operations, making it a critical factor in quantum chip design (Lu et al., 2012). However, this parameter must be carefully controlled within the constraints of coupling energy to avoid issues such as noise, cross-talk between qubits, and poor gate fidelity (Kwon et al., 2021).

The objective of our optimization problem is to maximize the coupling strength while adhering to the constraint imposed by the coupling energy. We aim to find the optimal configuration of the quantum chip that achieves the highest coupling strength possible within the predefined bounds of the energy constraint. To achieve this, we use a dataset comprising 393 data points, each describing the geometric and electrical features of the quantum chip. These features are utilized to calculate the energy associated with each chip configuration, including the coupling strength. (Li and Jin, 2023).

Generating this dataset is practical because it captures the variability in quantum chip designs, materials, and operational conditions. Furthermore, it allows us to explore and optimize the coupling strength systematically, which is crucial for improving the performance and efficiency of quantum computations (Liu et al., 2022; Miller, 1997). The choice of the energy feature, denoted as $E q_1 q_2$, as the target for constraint optimization is based on experimental insights and considerations, which may vary for different quantum chips (Li and Jin, 2023). By maintaining precise control over this energy feature within the predefined range, we demonstrate the effectiveness of our proposed optimization method in achieving our objectives in the context of synthetic quantum chip design.

### E.4 DATASET COLLECTION

We initiate our experiment by curating a comprehensive dataset comprising 393 data points. Each data point encompasses eleven geometric features describing the physical dimensions and layout of quantum chip components. These are complemented by four electrical features obtained through rigorous simulations utilizing standard electrical simulation software. The energy associated with each quantum chip configuration is derived from these four electrical features. Additionally, we calculate the coupling strength for each specific quantum chip layout using the energy and reference frequency.

### E.5 MODEL ARCHITECTURE

In quantum chip design, the coupling strength between qubits is a critical parameter. It determines the rate at which qubits can exchange quantum information, and thus influences the speed and fidelity of quantum operations (Lu et al., 2012).

However, the coupling strength must be carefully controlled under the constraints of coupling energy. If the coupling energy is too high (i.e., the qubits are too strongly coupled), it can lead to unwanted effects such as noise or cross-talk between qubits (Kwon et al., 2021). Cross-talk is a phenomenon where a signal transmitted on one qubit influences another qubit, leading to errors in quantum operations (Sarovar et al., 2020).

On the other hand, if the coupling energy is too low (i.e., the qubits are weakly coupled), it can result in poor gate fidelity (Ghosh and Geller, 2010). Gate fidelity is a measure of how accurately quantum gates (the basic operations of a quantum computer) can be implemented. If the gate fidelity is low, the output of a quantum computation may be unreliable (Ghosh and Geller, 2010).

Therefore, in the context of quantum chip design, achieving a higher coupling strength within a certain constraint of coupling energy is often desirable. It allows for fast and accurate quantum operations while avoiding the problems associated with too much or too little coupling.

Precise control over the energy feature within specific bounds is pivotal in optimizing the coupling strength during the quantum chip design process for several reasons:

1. **Optimal Coupling Strength**: The coupling strength between qubits in a superconducting quantum chip is intricately linked to the system's energy feature. Meticulous management of these parameters, closely tied to the energy feature, enables the attainment of optimal coupling strength. This stands as a crucial factor in ensuring the high fidelity of two-qubit gates (Li and Jin, 2023).

2. **Energy Efficiency**: Devices such as superconducting diodes can achieve enhanced energy efficiency when equipped with a series of gates to control the energy flow (Gupta et al., 2023)[3].

3. **Suppression of Energy Loss Channels**: The choice of material and geometric design of the sample plays a pivotal role in minimizing qubit energy loss channels[4] (Lienhard et al., 2019).

The constraints associated with the energy feature in a quantum chip hold significant sway over the performance and viability of quantum computations. These constraints exhibit variability based on the specific quantum chip under consideration, owing to differences in design, materials, and operational conditions (Yang et al., 2020; Hao et al., 2022; Kwon et al., 2021)

For instance, most quantum computers under global development only operate at fractions of a degree above absolute zero, necessitating multi-million-dollar refrigeration (Yang et al., 2020). Nevertheless, researchers have pioneered a proof-of-concept quantum processor unit cell that functions at 1.5 Kelvin, which is 15 times warmer than competing chip-based technologies (Yang et al., 2020). This elevated operating temperature holds promise for more cost-effective and robust quantum computers (Yang et al., 2020).

Furthermore, the incorporation of complex constraints poses a central challenge when applying near-term quantum optimization algorithms to industrially relevant problems (Hao et al., 2022). In general, such constraints cannot be easily encoded in the circuit, and there is no guarantee that quantum circuit measurement outcomes will adhere to these constraints (Hao et al., 2022). Consequently, novel approaches for solving constrained optimization problems with unconstrained, readily implementable quantum ansatze are being proposed (Hao et al., 2022).

However, it's imperative to acknowledge that the appropriate constraints for the energy feature can fluctuate depending on the specific quantum chip under scrutiny. These constraints may necessitate experimental observations for determination, with their range often rooted in prior experimental insights. This is because the energy feature's range can exert a substantial influence on the performance and feasibility of quantum computations (Liu et al., 2022; Miller, 1997).

In our synthetic quantum chip example, we opted to focus on one specific energy feature, denoted as $Eq_1q_2$, as the target for constraint optimization. In practical applications, the range for this energy feature should be selected based on prior experimental insights and considerations, which may vary for each unique chip. Nevertheless, for our synthetic chip, we delineated the range for this energy feature ourselves, constraining it within the bounds of $30 \times 10^{-23}$J to $45 \times 10^{-23}$J.

By maintaining precise control over the energy feature within this pre-defined range, we systematically explored and optimized the coupling strength for the synthetic superconducting quantum chip. This experiment serves as a compelling demonstration of the efficacy of our proposed method in achieving our optimization objectives.

---

[3]This feature has not been integrated into superconducting diodes previously.

[4]Qubits can spontaneously dissipate energy through dielectric defects on the surface and interfaces of the sample or by coupling to unwanted package modes.

### E.6 EXTRACTION OF FEATURES AND COMPUTATION

In this study, we extract a set of 11 geometric features that encompass various aspects of the quantum chip, such as parameters related to its length, width, the gap between components (qubit and coupler), and the boundary layer. Following electrical simulations, we obtain four crucial electrical values: the self-capacitance of the two qubits ($C_{01}$ and $C_{02}$), the self-capacitance of the coupler ($C_{0c}$), and the mutual capacitance between each qubit ($C_{12}$) and between each qubit and the coupler ($C_{1c}$ and $C_{2c}$). For this structural context, we make the assumption of identical and symmetric properties for the two qubits, resulting in $C_{1c} = C_{2c}$ and $C_{01} = C_{02}$.

The energy associated with each configuration of the quantum chip is computed using the following equations (Sete et al., 2021; Li and Jin, 2023):

$$E_{Cq_1} = \frac{e^2}{2\left(C_{01} + C_{1c} + C_{12}\right)},$$
$$E_{Cc} = \frac{e^2}{2\left(C_{0c} + C_{1c} + C_{2c}\right)},$$
$$E_{Cq_2} = \frac{e^2}{2\left(C_{02} + C_{2c} + C_{12}\right)},$$
$$E_{q_1 c} = e^2\left(-C_{1c}\right)^{-1},$$
$$E_{q_2 c} = e^2\left(-C_{2c}\right)^{-1},$$
$$E_{q_1 q_2} = e^2\left(-C_{12}\right)^{-1},$$

where $e$ represents the elementary charge in coulombs (C). For convenience in calculation, we adopt $e = 1.602$ (rather than $1.602 \times 10^{-19}$). The corresponding coupling strength $g$ is given by (Sete et al., 2021; Li and Jin, 2023):

$$g = \frac{2\omega_q}{B}\left(A - \frac{\omega_c^2}{\omega_c^2 - \omega_q^2}\right),$$

where

$$A = \frac{2E_{12}E_{Cc}}{E_{1c}E_{2c}}, \quad B = \frac{16E_{Cq}E_{Cc}}{E_{1c}E_{2c}},$$

and $\omega_c$ and $\omega_q$ represent the frequencies of the coupler and qubits, respectively. These values are contingent on experimental results and may vary across different studies. According to (Li and Jin, 2023), $\omega_c$ ranges from 9 GHz to over 16 GHz, while for the Zuchongzhi 2.1 quantum chip, $\omega_q$ is 5.099 GHz, and for Sycamore, it is 6.924 GHz. In our synthetic chip, we set $\omega_c = 20$ GHz and $\omega_q = 6.9$ GHz.

These equations encapsulate the energy associated with each component of the quantum chip configuration. The computed energy values furnish valuable insights into the performance and efficiency of the quantum chip.

### E.7 REGRETS DUE TO SUBOPTIMALITY AND CONSTRAINT VIOLATION

We plot the regrets at the estimator $\tilde{\mathbf{x}}_t$ resulting from suboptimality, i.e., $r_f(\tilde{\mathbf{x}}_t)$, and from constraint violation, i.e., $\sum_{c \in \mathcal{C}} r_c(\tilde{\mathbf{x}}_t)$, against the number of input queries in Fig. 4. The use of a log-scale causes the line plot to extend beyond the plotting area when the regret is exactly 0 (a log value of $-\infty$).

Fig. 4a shows that the estimator $\tilde{x}_*$ of the optimal solution is typically a feasible input when there are no active constraints at the optimal solution (such as in the problem [S-A0]). However, in cases where active constraints exist at the optimal solution, accurately estimating the boundary of the feasible region at the optimal solution becomes challenging. Consequently, the estimator often becomes an

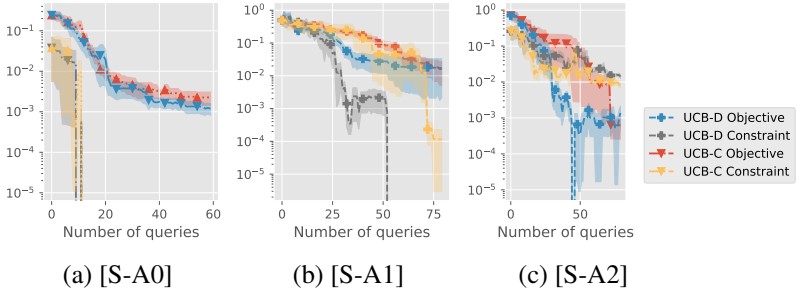

(a) [S-A0]        (b) [S-A1]        (c) [S-A2]

Figure 4: Plot of the regret $r_f(\tilde{\mathbf{x}}_t)$ at the estimator due to suboptimality (labeled as *UCB-C Objective* and *UCB-D Objective*) and the regret $\sum_{c \in \mathcal{C}} r_c(\tilde{\mathbf{x}}_t)$ due to constraint violation (labeled as *UCB-C Constraint* and *UCB-D Constraint*) against the number of input queries.

infeasible input (Figs. 4b,c). Nevertheless, it is worth noting that the sum of regrets serves as an upper bound for the regrets associated with the objective function and constraint functions. Hence, even if the estimator is infeasible, the no-regret result in Lemma 3.5 shows that we can achieve arbitrarily small constraint violation at the expense of more BO iterations.

Appendix A delves into the rationale behind our selecting an estimator within the optimistic feasible region, even if it may be infeasible, particularly in the context of equality constraints. More generally, if the feasible region has an empty interior (e.g., due to equality constraints), pinpointing a feasible input accurately becomes impossible, regardless of the number of observations gathered. For example, if the feasible region is only a line in the space, then estimating the line without any error (to identify feasible inputs) using only noisy observations is not possible. Our strategy of utilizing the optimistic feasible region circumvents this challenge, albeit with a minor constraint violation if active constraints are present at the optimal solution. It is important to recall that in situations where there are no active constraints at the optimal solution, our estimator is often a feasible input, as illustrated in Fig. 4a. Additionally, as the number of BO iterations increases, the amount of constraint violation approaches zero, allowing for arbitrarily small violations at the expense of a more extended BO procedure.