# OpenReview forum: "Optimistic Bayesian Optimization with Unknown Constraints"
_ICLR.cc/2024/Conference — ICLR 2024 poster_

### Official Review · Reviewer_WGLv · 2023-10-29

**Soundness:** 2 fair
**Presentation:** 3 good
**Contribution:** 2 fair
**Rating:** 6
**Confidence:** 4

**Summary:**

This paper proposes a constrained Bayesian optimization algorithm which aims at dealing with constrained black-box optimization under decoupled setting. The authors utilized confidence bound derived from Gaussian process to determine the function oracle to query, and derived the cumulative regret bound in terms of both maximization and summation computation way. Experiment on synthetic function and real-world application demonstrates the query-efficiency of the proposed UCB-D algorithm.

**Strengths:**

1. The idea is clear and easy to follow.

2. The paper is well-written, and the presentation from the active learning aspect helps better understanding the proposed algorithm.

3. The experiment result well demonstrates the query efficiency of UCB-D.

**Weaknesses:**

1. The paper does not explicitly introduce the motivation of decoupling the function query, and seems that the chosen real-world benchmark does not has the property of decoupling the function queries.

**Questions:**

1. As mentioned in weakness part, my major concern lies in the motivation of decoupling the function queries, since in many real-world applications, the objective and constraint values are simultaneously evaluated after one trial, and decoupling the function query seems not save the cost. Can you explain the motivation and potential application of decoupling the function queries?

2. The result plot only shows the summation of regrets, which does not tell the found solution is feasible or not. Can you separately show the regret of  the objective function and constrant functions?

3. In Figure 2 (h), the standard error of UCB-D is much larger than other baselines. Can you give some insights of why this happens?

---

> ### Author Response · Authors · 2023-11-19
> **Authors Response (Part 1)**
>
> We appreciate your valuable feedback and recognition of the clarity and ease of understanding in our idea, presentation, and experimental results. We will seriously consider your feedback during the revision of our paper.
>
> We would like to address your questions below:
>
> > 1. As mentioned in weakness part, my major concern lies in the motivation of decoupling the function queries, since in many real-world applications, the objective and constraint values are simultaneously evaluated after one trial, and decoupling the function query seems not save the cost. Can you explain the motivation and potential application of decoupling the function queries?
>
> We would like to emphasize that our methodology and theoretical analysis are applicable to both coupled and decoupled queries, as mentioned in the final paragraph of Sec. 1: Introduction. Therefore, our research holds significance not only for the decoupled setting but also for the coupled setting of constrained BO. We will further enhance the motivation and potential application of decoupling the function queries in our revised paper as follows:
>
> **Motivation:** In Remark 3.2, we elaborate on the motivation that decoupling the function queries results in a more query-efficient solution. Hence, leveraging the ability to evaluate the objective function and constraints independently becomes advantageous, particularly in situations where the evaluation of each component is resource-intensive. For instance, when there is a high probability that the input query is feasible, the evaluation of constraints at that input query becomes unnecessary (especially if the optimal solution is far from the boundary of the feasible region, as illustrated in Figure 2a). In such scenarios, significant cost savings can be realized in the decoupled setting compared to the coupled one.
>
> **Potential applications:** Here are two potential applications that showcase the benefits of employing decoupled queries:
>
>    + A practical scenario of decoupled queries arises in the automotive industry. When designing the body of a car, various approaches are employed to assess its performance. For instance, computational fluid dynamics (CFD) is utilized to model the interaction between the car body and air flows. On the other hand, finite element analysis (FEA) is employed to simulate the car body's response to various loads, such as those encountered in collisions. One may construct a constrained optimization problem to maximize the aerodynamic performance of a car while adhering to safety constraints during collisions. CFD and FEA can be performed independently and performing each of them is expensive.
>
>    + Another situation arises when selecting an ML model for deployment across different edge devices. These devices have diverse hardware configurations, leading to variations in prediction times and battery consumption. The goal is to optimize the model's prediction accuracy while meeting constraints on prediction times and battery consumption for various edge devices. Evaluating the prediction time and battery consumption on different devices and the prediction accuracy of the ML model can be performed separately, and these evaluations are time-consuming.

---

> ### Author Response · Authors · 2023-11-19
> **Authors Response (Part 2)**
>
> > 2. The result plot only shows the summation of regrets, which does not tell the found solution is feasible or not. Can you separately show the regret of the objective function and constrant functions?
>
>    + We have implemented the reviewer's recommendation by including regret plots for both the objective function and constraint functions separately in Appendix E.7 in our latest revision.
>
>    + It is important to highlight that the estimator $\tilde{x}^*$ of the optimal solution is typically a feasible input when there are no active constraints at the optimal solution, as shown in Figure 4a in the newly-added Appendix E.7. However, in cases where active constraints exist at the optimal solution, accurately estimating the boundary of the feasible region becomes challenging. Consequently, the estimator often becomes an infeasible input (the newly-added Figures 4b & 4c in Appendix E.7). Nevertheless, it is worth noting that the sum of regrets serves as an upper bound for the regrets associated with the objective function and constraint functions. Hence, even if the estimator is infeasible, the no-regret result in Lemma 3.5 shows that we can achieve arbitrarily small constraint violation at the expense of more BO iterations.
>
>    + Appendix A discusses the rationale underlying our selection of an estimator within the optimistic feasible region, even if it may be infeasible, particularly in the context of equality constraints. More generally, if the feasible region has an empty interior (e.g., due to equality constraints), pinpointing a feasible input accurately becomes impossible, regardless of the number of observations gathered. For example, if the feasible region is only a line in the space, then estimating the line without any error (to identify feasible inputs) using only noisy observations is not possible. Our strategy of utilizing the optimistic feasible region circumvents this challenge, albeit with a minor constraint violation if active constraints are present at the optimal solution. It is important to recall that in situations where there are no active constraints at the optimal solution, our estimator is often a feasible input, as illustrated in Figure 4a. Additionally, as the number of BO iterations increases, the amount of constraint violation approaches zero, allowing for arbitrarily small violations at the expense of a more lengthy BO procedure.
>
>    + When desiring a feasible solution and knowing that the interior of the feasible region is non-empty, there are several ways:
>
>      + One approach involves tightening the constraints by adding a small positive value to $\lambda_c$ for all $c \in \mathcal{C}$. However, caution is warranted as this adjustment may render the optimization problem infeasible if the interior of the feasible region is empty.
>
>      + Alternatively, instead of relying on an optimistic feasible region, a pessimistic feasible region can be utilized to ensure the feasibility of all inputs within it. It is important to note that this approach is not viable for optimization problems with equality constraints, as it would render the problem infeasible.
>
> > 3. In Figure 2 (h), the standard error of UCB-D is much larger than other baselines. Can you give some insights of why this happens?
>
>    This is due to the logarithmic scale used in the plot, which accentuates small values. For example, on the log plot, the distance between 0.01 and 0.1 (a span of 0.09) is visually the same as the distance between 0.1 and 1.0 (a span of 0.9). Consequently, although the standard error of UCB-D appears larger in Figure 2h compared to other methods, it is indeed smaller than most of the methods. In Figure 2h, at the end of the line plot, the numerical value of the standard error for UCB-D is 0.015506. In comparison, the standard errors for ADMMBO, CMES-IBO, EIC, and UCB-C in Figure 2h are 0.37287908, 0.02479963, 0.01956778, and 0.01082636, respectively.
>
> We sincerely wish that our motivation to the decoupled queries, and the additional plots of the regrets associated with the objective function and the constraints can improve your opinion on our paper.

---

> ### Comment · Reviewer_WGLv · 2023-11-20
>
> Thank you for your response about the motivation of the problem setting and clarification of the experimental detail. I think I understand this paper clearer.
>
> However, I still think the chosen real-world experiment does not utilize the advantage of decoupled evaluation. I encourage the authors to design more suitable real-world experiments based on the aforementioned potential applications to better demonstrate the advantage of UCB-D.
>
> Overall, I've increased my score from 5 to 6.

---

> > ### Author Response · Authors · 2023-11-21
> > **Grateful for Your Quick Response and The Score Update**
> >
> > We are glad that our response has enhanced your understanding and perspective regarding our paper. Thank you very much for your feedback and the improvement in the review score.
> >
> > As an additional remark, we realize that our CNN experiment can be approached from the decoupled setting when the validation datasets are owned by distinct data owners. This happens when various data owners wish to work together to train a shared model but prefer to keep their confidential validation sets private. In this experiment, we address the challenge of maximizing the average accuracy across all classes while ensuring that the recall of each class is at least $0.5$. By assuming the recall of each class is measured on validation sets owned by different data owners, separate from the validation set used to assess overall accuracy across all classes, the querying of constraints becomes decoupled from each other and from the objective function.

---

### Official Review · Reviewer_xiLP · 2023-10-31

**Soundness:** 4 excellent
**Presentation:** 3 good
**Contribution:** 4 excellent
**Rating:** 8
**Confidence:** 3

**Summary:**

The paper addresses the challenge of constrained BO within a decoupled setting, where evaluations of the objective function and constraints occur independently at different inputs. The decoupled setting requires adaptive selection between evaluating the objective function or a constraint, alongside selecting an input.  Additionally, the paper empirically evaluates the performance of the proposed algorithms using both synthetic and real-world optimization problems.

**Strengths:**

By allowing separate evaluations, the decoupled setting mirrors a more practical and realistic scenario, offering flexibility in the optimization process. This approach is more adaptable to various situations and potentially reduces computational expenses or time. Additionally, the proposed method is equipped with a theoretically proven performance guarantee.

**Weaknesses:**

n/a

**Questions:**

1. In plotting the UCB-C's regret against the number of queries (e.g., Figure 2g), as evaluations are conducted at both the objective function and constraints in each BO iteration, how are they plotted? Does the plot advance by jumping every #constraints + 1?
2. Just curious, in cases where the query point is feasible, why don’t we evaluate the constraints together with the objective function? What are the advantages of strictly adhering to UCB-D?

---

> ### Author Response · Authors · 2023-11-19
> **Authors Response**
>
> We thank you for recognizing our efforts in presenting a solution that is adaptable to various situations and equipped with a theoretically proven performance guarantee. Your valuable feedback will be carefully considered as we revise our paper.
>
> We would like to provide clarifications to your questions below:
>
> > 1. In plotting the UCB-C's regret against the number of queries (e.g., Figure 2g), as evaluations are conducted at both the objective function and constraints in each BO iteration, how are they plotted? Does the plot advance by jumping every #constraints + 1?
>
> You are correct in noting that for UCB-C (involving coupled queries), the plot advances by jumping every #constraints + 1 queries. For UCB-D (involving decoupled queries), the plot advances with a jump every single query.
>
>
> > 2. Just curious, in cases where the query point is feasible, why don’t we evaluate the constraints together with the objective function? What are the advantages of strictly adhering to UCB-D?
>
> As our objective is to minimize the number of queries, we aim to minimize unnecessary queries to either the objective function or the constraints. Consequently, if we know that a query point is feasible, querying the constraints does not provide additional information on its feasibility. Therefore, we propose to query only the objective function to determine whether the query point is suboptimal (i.e., indicated by a low objective function evaluation). However, when the cost of querying constraints is not prohibitive or the queries are coupled, we may choose to evaluate both the constraints and the objective function simultaneously, as proposed in the UCB-C algorithm.
>
> We sincerely hope that the above clarifications will improve your favorable perspective of our paper.

---

> > ### Comment · Reviewer_xiLP · 2023-11-22
> >
> > Thank you for the clarification.

---

> > > ### Author Response · Authors · 2023-11-23
> > > **Thank You for Your Feedback and Reviewing**
> > >
> > > We wanted to express our sincere gratitude for your evaluation of our paper.
> > > It is our pleasure to respond to your questions to enhance the understanding of the paper.

---

### Official Review · Reviewer_TVyc · 2023-10-31

**Soundness:** 2 fair
**Presentation:** 3 good
**Contribution:** 2 fair
**Rating:** 5
**Confidence:** 4

**Summary:**

This paper studies the Bayesian optimization with unknown constraints. Different from previous work, this paper focuses on the decoupled setting where objective function and constraints are evaluated independently at different inputs. A new constrained BO algorithm is proposed, and empirical results show the effectiveness of the algorithm.

**Strengths:**

1. This paper studies the constrained Bayesian optimization problem in the decoupled setting, which was a problem seldom studied before.
2. The whole paper is well organized, and I like the illustration figures in Figure 1, which are helpful.
3. Experiments on both synthetic and real-world problems are conducted to show the effectiveness of proposed algorithm.

**Weaknesses:**

1. In Introduction, the motivation of studying decoupled constrained BO (CBO) is unclear to me. What are the real-world applications of decoupled CBO? In which case should we evaluate objective function and constraints at different inputs? In that case, can we run several independent standard BOs to solve all problems separately? Or can we run multi-objective BO to solve it?
2. I’m surprised to see definition of regret in eq (3) by combining objective function evaluation together with constraint violations. They may sit in totally different function ranges. Let F denote the range of objective function and let C denote the range of constraints. If C is much greater than F, then regret has little information about convergence. Also, the optimal point x* is defined w.r.t. objective function and $r_c$ is independent to x*. Why is $r_c$ is part of the regret?
3. How do you solve optimization problems in Line 3 and 4 in Algorithm 1? Definition of $O_t$ seems like making solving Line 3 intractable.

**Questions:**

1. How does the last equation hold in eq 8 by adding vertical exploration bonus?
2. In last paragraph of Section 3.1, how does Lemma 3.1 imply that $O_t$ is non-empty with probability $>1-\delta$?

---

> ### Author Response · Authors · 2023-11-19
> **Authors Response (Part 1)**
>
> We thank you for appreciating our contributions (i.e., tackling a problem seldom studied before, a well-organized paper with helpful illustrations, experiments on synthetic and real-world problems showing the effectiveness of our algorithm) and providing useful feedback which we would take into account seriously when revising our paper.
>
> We would like to address your concerns below:
>
> > 1. In Introduction, the motivation of studying decoupled constrained BO (CBO) is unclear to me. What are the real-world applications of decoupled CBO? In which case should we evaluate objective function and constraints at different inputs? In that case, can we run several independent standard BOs to solve all problems separately? Or can we run multi-objective BO to solve it?
>
>    **Unified approach to both coupled and decoupled queries:** We would like to highlight that our approach and theoretical analysis work for both coupled and decoupled queries (see last paragraph of Sec. 1: Introduction). Thus, our work is important to not only the decoupled setting but also the coupled setting of constrained BO.
>
>    **In which case should we evaluate objective function and constraints at different inputs?**
>    We should evaluate objective function and constraints at different inputs when we can evaluate the objective function and constraints independently and the evaluation of each of them is expensive. In Remark 3.2, we also explored the circumstances under which employing decoupled queries could significantly improve efficiency in terms of the number of queries, as compared to coupled queries. For instance, when there is a high probability that the input query is feasible, the evaluation of constraints at that input query becomes unnecessary (especially if the optimal solution is far from the boundary of the feasible region, as illustrated in Figure 2a). In such scenarios, significant cost savings can be realized in the decoupled setting compared to the coupled one.
>
>    **Real-world applications of decoupled queries:**
>
>    + A practical scenario of decoupled queries arises in the automotive industry. When designing the body of a car, various approaches are employed to assess its performance. For instance, computational fluid dynamics (CFD) is utilized to model the interaction between the car body and air flows. On the other hand, finite element analysis (FEA) is employed to simulate the car body's response to various loads, such as those encountered in collisions. One may construct a constrained optimization problem to maximize the aerodynamic performance of a car while adhering to safety constraints during collisions. CFD and FEA can be performed independently and performing each of them is expensive.
>
>    + Another situation arises when selecting an ML model for deployment across different edge devices. These devices have diverse hardware configurations, leading to variations in prediction times and battery consumption. The goal is to optimize the model's prediction accuracy while meeting constraints on prediction times and battery consumption for various edge devices. Evaluating the prediction time and battery consumption on different devices and the prediction accuracy of the ML model can be performed separately, and these evaluations are time-consuming.
>
>    We will incorporate the above examples into Sec. 1: Introduction to further motivate the decoupled queries.
>
> > In that case, can we run several independent standard BOs to solve all problems separately? Or can we run multi-objective BO to solve it?
>
>    + We think that performing several independent standard BOs to solve a constrained BO problem with decoupled queries is a solution that is orthogonal to all existing constrained BO solutions. However, based on our current understanding, it is unclear (at least to us) how this can be accomplished effectively.
>
>    + Similarly, it is unclear to us how multi-objective BO can be used to solve constrained BO if the queries are decoupled. We would appreciate the reviewer's guidance in providing further elucidation on these two approaches.

---

> > ### Comment · Reviewer_TVyc · 2023-11-22
> >
> > Thanks for your feedback and real-world application examples, which are helpful. Previously, "evaluating the objective function and constraints independently at different points" makes me confused because it seems that you are solving a objective function optimization problem and other constraint function optimization problems at the same time rather than solving a CBO problem. That's why I asked the question if we can run several independent BOs or run a multi-obj BO. I appreciate that you will incorporate these examples into the next version.

---

> ### Author Response · Authors · 2023-11-19
> **Authors Response (Part 2)**
>
> > 2. I'm surprised to see definition of regret in eq (3) by combining objective function evaluation together with constraint violations. They may sit in totally different function ranges. Let $F$ denote the range of objective function and let $C$ denote the range of constraints. If $C$ is much greater than $F$, then regret has little information about convergence. Also, the optimal point $x^\ast$ is defined w.r.t. objective function and $r_c$ is independent to $x^\ast$. Why is $r_c$ part of the regret?
>
>
>    + **Why $r_c$ is part of the regret:**
>
>      > Also, the optimal point $x^\ast$ is defined w.r.t. objective function and $r_c$ is independent to $x^\ast$. Why is $r_c$ part of the regret?
>
>      It is important to clarify that the determination of $x^\ast$ is **not solely** based on the objective function in our context, as we are dealing with a constrained optimization problem. Consequently, $x^\ast$ is defined by considering both the objective function and the constraints, seeking to maximize the former while adhering to the latter. Therefore, the incorporation of $r_c$ (which quantifies the degree of constraint violation for a given input) into the regret is crucial. The regret, in this context, serves to evaluate the performance of an estimator $\tilde{x}_t$ in approximating the optimal solution. As a result, this regret must account for both suboptimalities arising from the objective function and the violation of constraints. Hence, our proposed regret that addresses these aspects is a unique contribution of our work, particularly considering the limited existing literature on regret analysis in constrained BO.
>
>    + **Different function ranges**
>
>      > I'm surprised to see definition of regret in eq (3) by combining objective function evaluation together with constraint violations. They may sit in totally different function ranges. Let $F$ denote the range of objective function and let $C$ denote the range of constraints. If $C$ is much greater than $F$, then regret has little information about convergence.
>
>      + From a theoretical analysis perspective, the no-regret definition exhibits asymptotic behavior, specifically, the simple regret tends to $0$ as the number of iterations $T$ approaches infinity. The impact on convergence as $T$ approaches infinity remains unaffected by the magnitude of the function range, provided that the function range is not unbounded. Our proposed regret serves as an upper bound for regrets arising from the objective function and constraints, as it represents the maximum or sum of these regrets. Consequently, if our proposed regret approaches $0$, it implies that both the regrets associated with the objective function and constraints must also approach $0$, even in cases where their function ranges may differ. This can be conceptualized as constraining the worst-case regret among all regrets incurred by the objective function and constraints. We would like to point out that in the classic GP-UCB work, the asymptotic convergence holds irrespective of the scale of the function range.
>
>      + From a practical point of view, one can normalize the observations at every BO iteration so that the observations from the objective function and constraints belong to similar ranges.
>
> ---
>
> > 3. How do you solve optimization problems in Line 3 and 4 in Algorithm 1? Definition of $\mathcal{O}_t$ seems like making solving Line 3 intractable.
>
>    + Line 3: This is a constrained optimization problem with a closed-form expression of the objective function and closed-form expressions of the constraints. Hence, it can be solved with any existing constrained optimization package.
>      + Objective function: $u_{f,t-1}(x)$;
>      + Constraints: $u_{c,t-1}(x) \ge \lambda_c \ \ \forall c \in \mathcal{C}$ (from the definition of $\mathcal{O}_t$ in equation 9);
>      + As defined in equation 7, $u_{h,t-1}$ only consists of the GP posterior mean and GP posterior standard deviation, both of which have closed-form expressions for a Gaussian process model. The set $\mathcal{C}$ is a finite set of constraints.
>
>    + Line 4: We simply go through all constraints and find the one with the largest $\lambda_c - l_{c,t-1}(x_t)$.

---

> > ### Comment · Reviewer_TVyc · 2023-11-22
> >
> > I agree that your new definition of regret is the upper bound of regrets arising from the objective function and constraints. However, I still have concern that this definition shows too little information about the objective function optimization. Without working with any specific algorithm, let's imagine regret of objective function is really large but those of constraint functions are relatively low, then we are not really solving a CBO problem but trying hard to fit the constraints. Also, even in practice, if they are in different function ranges, how does the algorithm to normalize the observations to similar function ranges without knowing them in advance?

---

> > > ### Author Response · Authors · 2023-11-22
> > > **Motivation for the First Regret Notion in CBO for Decoupled Queries**
> > >
> > > We would like to address your question about the regret notion as follows.
> > >
> > > > I agree that your new definition of regret is the upper bound of regrets arising from the objective function and constraints. However, I still have concern that this definition shows too little information about the objective function optimization. Without working with any specific algorithm, let's imagine regret of objective function is really large but those of constraint functions are relatively low, then we are not really solving a CBO problem but trying hard to fit the constraints. Also, even in practice, if they are in different function ranges, how does the algorithm to normalize the observations to similar function ranges without knowing them in advance?
> > >
> > > It is important to highlight that our utmost aim is the minimization of regret. Hence, our greater focus is on the ultimate outcome of the **entire** optimization process (i.e., asymptotically no-regret) rather than concentrating only on a particular iteration (especially when such an iteration occurs early in the CBO procedure yielding a large regret or unsatisfactory result). This is also the gist behind the asymptotic no-regret analysis (i.e., limiting behavior) which is employed in the seminal GP-UCB work of Srinivas et al. (2010).
> > >
> > > That being said, considering **the example provided by the reviewer** (i.e., the regret of objective function is really large but those of constraint functions are relatively low), there are $2$ cases:
> > >
> > > + Supposing the maximum of regrets is upper bounded by, say, 0.001 (even though the regret of objective function is still relatively larger than that of the constraint functions), we believe we have solved the CBO problem with a specific level of accuracy. According to our regret analysis, this should happen after a number of BO iterations.
> > >
> > > + On the other hand, supposing the maximum regret is upper bounded by, say, 1000, this indicates that the result is unsatisfactory. So, one who prefers a better objective function evaluation should continue running the algorithm. Considering our algorithm, it is most likely that the objective function will be chosen for querying, given its higher regret. Consequently, the objective function will not be starved of attention, indicating that our algorithm is not solely trying very hard to fit the constraints.
> > >
> > > Moreover, as highlighted in our earlier response, "This can be conceptualized as constraining the worst-case regret among all regrets incurred by the objective function and constraints". It is important to emphasize that in **robust optimization**, a prevalent paradigm is to minimize (or maximize) the worst-case scenario where the function range may vary across different scenarios.
> > >
> > > Importantly, **in Remark 2.1 of the original paper, we have also explored the relationship between formulating regret as a sum of regrets (of both the objective function and constraints) vs. expressing it as the maximum of regrets**; in the former case, the regrets incurred by the constraints, regardless of their magnitude, would not "subsume" that of the objective function. Interestingly, these two regret notions are essentially equivalent in the context of no-regret analysis because $r(x) \le s(x) \le |\mathcal{F}| r(x)$ and $s(x) / |\mathcal{F}| \le r(x) \le s(x)$ where $r(x)$ and $s(x)$ represent the maximum and sum of regrets, respectively. Consequently, our theoretical analysis is applicable to both forms of regret notions. We wish to also note that in the context of batch BO, the practice of defining regret as the sum of regrets (over inputs in the batch) and considering the best-case regret is not uncommon, as seen in [1,2].
> > >
> > > Finally, it is worth mentioning that to the best of our knowledge, **we have not seen a regret notion (and even more so, a no-regret analysis) being established for CBO with decoupled queries**. We consider it a **significant contribution to introduce a regret notion that is amenable to both a proof of sublinear cumulative regret and the development of an algorithm that demonstrates empirical effectiveness**.
> > >
> > > To further improve our understanding of the regret, we are interested to learn, at your convenience, about your perspective on the ideal regret notion in the context of the CBO problem. We also wish to appeal to you in considering an alternative perspective of the regret notions like ours in this work.
> > >
> > > References:
> > >
> > > [1] Gaussian process optimization with upper confidence boundand pure exploration. ECML/PKDD 2013.
> > >
> > > [2] Distributed batch Gaussian process optimization. ICML 2017.

---

> > > > ### Author Response · Authors · 2023-11-22
> > > > **On Practical Implementation with Normalization**
> > > >
> > > > Regarding the **normalization operation from a practical implementation standpoint**, we believe that while the functions are black-box (lacking a closed-form expression or derivatives), experts often have domain knowledge about the range of functions. For instance, experts might understand the range of impact force on a car based on the collision speed and the weights. It is also common for the accuracy of a machine learning model to be confined within the range [0,1]. In the rare cases where knowledge about the function range is absent, one can resort to normalizing all observations according to the range observed thus far (the maximum observation and the minimum observation so far). This resembles the process of normalizing features within a training dataset, where individuals may normalize features based on the maximum and minimum values of each feature in the (observed) dataset.
> > > >
> > > > We would include the above discussion in our revised paper. We sincerely hope that the above clarifications would help you better appreciate our work and improve your opinion.

---

> > > > ### Comment · Reviewer_TVyc · 2023-11-22
> > > >
> > > > Thanks for your clarification, which partially addressed my concern. Ideal regret notion of CBO is still in the air and I appreciate your work in making an attempt. I've increased my rating from 3 to 5.

---

> > > > > ### Author Response · Authors · 2023-11-23
> > > > > **Thank You for Your Valuable Feedback and Improved Score**
> > > > >
> > > > > We are grateful for your valuable feedback and the improved score, and we appreciate the ongoing discussion. While the time for further discussion is limited, we are open to providing any additional clarifications necessary to adequately address your concerns regarding the regret notion. The discussion so far has been instrumental in revealing that our regret notion can find motivation beyond the CBO context, from areas such as robust optimization and batch BO. We hope that our first attempt to address the decoupled setting with an intuitive implementation and theoretical performance guarantee will serve as inspiration for subsequent works in the field.

---

> ### Author Response · Authors · 2023-11-19
> **Authors Response (Part 3)**
>
> Further, we would like to address your questions below:
>
> > 1. How does the last equation hold in eq 8 by adding vertical exploration bonus?
>
> Equation 8 is directly adopted (without any modification) from the work of Srinivas et al. (2010). The term in the argmax is just the definition of the upper confidence bound of $f(x)$. The introduction of the name "vertical exploration bonus" for a term in equation 8 aims to emphasize the analogous concept of horizontal exploration, which becomes relevant in the context of constrained BO later on.
>
> ---
> > 2. In last paragraph of Section 3.1, how does Lemma 3.1 imply that $\mathcal{O}_t$ is non-empty with probability $> 1-\delta$?
>
> If the optimization problem is feasible, then there exists at least $1$ feasible input $x_0 \in \mathcal{X}$, i.e., for all $c \in \mathcal{C}$, $c(x_0) \ge \lambda_c$. Lemma 3.1 says that $u_{c,t-1}(x) \ge c(x)$ for all $x \in \mathcal{X}$, $t \ge 1$, and $c \in \mathcal{C}$ with probability $\ge 1 - \delta$. Hence, if the optimization problem is feasible, $u_{c,t-1}(x_0) \ge c(x) \ge \lambda_c$ for all $c \in \mathcal{C}$ and $t \ge 1$ with probability $\ge 1 - \delta$. In other words, $\mathcal{O}_t$ is non-empty with probability $\ge 1 - \delta$ for all $t \ge 1$. We will include this argument in the revised paper.
>
>
> We sincerely hope that our responses above have contributed positively to your opinion of our work in this paper. We are open to addressing any remaining concerns and providing further clarification.

---

> > ### Comment · Reviewer_TVyc · 2023-11-22
> >
> > Thanks for the feedback to point 2. However, for point 1, my questions is: let $x_1=\argmax_{x \in \mathcal{X}} \mu(x)$ and $x_2=\argmax_{x \in \mathcal{X}} \mu(x) + \beta^{1/2} \sigma(x)$, why $x_1=x_2$ in eq 8?

---

> > > ### Author Response · Authors · 2023-11-22
> > > **A Possibility of Misinterpreting Equation 8**
> > >
> > > Thank you for your response and follow-up questions. We would like to address your question about eq 8 as follows.
> > >
> > > > Thanks for the feedback to point 2. However, for point 1, my questions is: let $x_1 = \text{arg}\max_{x \in \mathcal{X}} \mu(x)$ and $x_2 = \text{arg}\max_{x \in \mathcal{X}} \mu(x) + \beta^{1/2}\sigma(x)$, why $x_1=x_2$ in eq 8?
> > >
> > > Eq. 8 in our paper is written as follows:
> > >
> > > $\arg\max_{x \in \mathcal{X}} u_{f,t-1}(x) = \arg\max_{x \in \mathcal{X}} (\mu_{f,t-1}(x) + \beta_t^{1/2} \sigma_{f,t-1}(x))$
> > >
> > > The term on the RHS after argmax is just the definition of **$u_{f,t-1}(x)$ (not $\mu(x)$)** on the LHS; this is  taken from the work of Srinivas et al. (2010). You may have misinterpreted **$u_{f,t-1}$** as **$\mu$**.

---

> > > > ### Comment · Reviewer_TVyc · 2023-11-22
> > > >
> > > > Good catch. Yes, I misinterpreted eq 8.

---

> ### Author Response · Authors · 2023-11-22
> **Seeking Your Valued Feedback on Our Response**
>
> We hope this message finds you well. We are following up on our recent response to your questions and reviews of our paper and would like to know whether our response adequately addressed all your concerns. Your feedback is crucial to us, so could you kindly take a moment to confirm if our rebuttal meets your expectations? We greatly appreciate the time and effort you have devoted to responding to us.

---

### Official Review · Reviewer_L67a · 2023-11-01

**Soundness:** 3 good
**Presentation:** 3 good
**Contribution:** 3 good
**Rating:** 8
**Confidence:** 4

**Summary:**

The paper proposes an algorithm tackling BO with unknown constraints by (1) explicitly measuring the benefits of querying the objective and the benefits of querying the constraint(s), (2) maximizing the general benefits to achieve an efficient trade-off of learning the unknown constraint and optimizing the unknown objective. The theoretical analysis offers a convergence guarantee of the proposed CBO method, a significant advancement in the domain.

**Strengths:**

1. The key concepts and proposed algorithm mostly rely on the confidence interval, which bears good interpretability.

2. The analysis extends GP-UCB results into the CBO setting.

3. The figures are illustrative, and the paper is, in general, well-organized.

**Weaknesses:**

1. Though the author highlights the connection of the proposed method to active learning (AL), it lacks a discussion on the link to the existing AL methods. For example, the concepts, including the uncharted area and $\nu_t$-relaxed feasible confidence region, resonate with the concepts in [1], and the analysis also bears connections.

2. The definition of regret is unconventional and lacks sufficient discussion. Typically, in the CBO setting, the reward is only defined within the feasible region, as there is no reward incurred by querying the points that are infeasible. The regret here is defined on both the objective and constraints, which circumvent the problem of infinite instantaneous regret in cumulative regret analysis when querying points out of the feasible region.

**References**
[1] Gotovos, Alkis. "Active learning for level set estimation." Master's thesis, Eidgenössische Technische Hochschule Zürich, Department of Computer Science,, 2013.

**Questions:**

1. Could the author include the line of work in constraint active search [2][3] in a discussion of related work? They are closely related to the CBO problem as it aims at searching feasible points efficiently within feasible regions defined by unknown constraints.

2. There is a recent paper studying a similar CBO solution [4] to the proposed algorithm. It is unnecessary to include it in the paper due to its release publication time, but I encourage the author to explore it.

**References**

[2] Malkomes, G., Cheng, B., Lee, E. H., & Mccourt, M. (2021, July). Beyond the pareto efficient frontier: Constraint active search for multiobjective experimental design. In International Conference on Machine Learning (pp. 7423-7434). PMLR.

[3] Komiyama, J., Malkomes, G., Cheng, B., & McCourt, M. (2022). Bridging Offline and Online Experimentation: Constraint Active Search for Deployed Performance Optimization. Transactions on Machine Learning Research.

[4] Zhang, F., Zhu, Z., & Chen, Y. (2023). Constrained Bayesian Optimization with Adaptive Active Learning of Unknown Constraints. arXiv [Cs.LG]. Retrieved from http://arxiv.org/abs/2310.08751

---

> ### Author Response · Authors · 2023-11-19
> **Authors Response (Part 1)**
>
> We appreciate your valuable reviews and thank you for appreciating the good interpretability of our key concepts and algorithm, our theoretical analysis of CBO, and our well-organized paper with illustrative figures. We will carefully incorporate your feedback and suggested related works into our revised paper.
>
> We would like to address your concerns below:
>
> > 1. Though the author highlights the connection of the proposed method to active learning (AL), it lacks a discussion on the link to the existing AL methods. For example, the concepts, including the uncharted area and $\nu_t$-relaxed feasible confidence region, resonate with the concepts in [1], and the analysis also bears connections.
>
>    We will include the discussion of [1] in the revised paper. The major differences with [1] are briefly discussed below.
>
>    + Our approach is not simply a combination of active learning for level set estimation and Bayesian optimization. As shown in Figs. 2a-c, our method adaptively allocates input queries based on specific scenarios, eliminating the necessity for performing level set estimation across the entire feasible region. In contrast, applying active learning for level set estimation typically involves uniformly exploring the complete feasible region.
>    + The work of [1] requires monotonically decreasing confidence region by intersecting successive confidence intervals. Practical applications may encounter challenges, particularly if the intersection results in an empty confidence region. In contrast, our approach aligns more closely with conventional GP-UCB method, omitting the requirement for such a monotonic decrease in the confidence region and thereby avoiding the issue of empty confidence regions.
>    + In [1], manually setting the value of $\epsilon$ is mainly for the theoretical analysis from our perspective. In our work, the value of $\nu_t$ is not manually set, but it is adaptively chosen by our algorithm. Furthermore, it is to balance between the vertical and horizontal exploration, a concept that does not exist in [1].
>
> > 2. The definition of regret is unconventional and lacks sufficient discussion. Typically, in the CBO setting, the reward is only defined within the feasible region, as there is no reward incurred by querying the points that are infeasible. The regret here is defined on both the objective and constraints, which circumvent the problem of infinite instantaneous regret in cumulative regret analysis when querying points out of the feasible region.
>
>    The conventional viewpoint is that a constrained minimization problem with known constraints is often formulated as an unconstrained minimization problem with infinite values for infeasible inputs. The constraints in constrained BO are unknown, and so is its feasible region. Assigning infinite instantaneous regret to infeasible inputs is problematic. For example, when there are equality constraints, it is impossible to estimate the feasible region exactly (e.g., a line) given noisy observations. In such a case, we will never know if an input is feasible. Hence, without a finite measure of constraint violation, analyzing the regret is very challenging. As a result, we believe it is a natural approach to assign finite regret based on constraint violation. We have discussed the challenges of equality constraints in Appendix A. We will incorporate this discussion in the revised paper.
>
>    Additionally, the limited theoretical works on constrained BO (only 1 theoretical work with coupled queries and no theoretical work with decoupled queries) deprive us of a conventional regret definition capable of addressing both the unknown objective function and unknown constraints. Hence, we consider introducing this concept of regret as a distinctive contribution of our work.

---

> > ### Author Response · Authors · 2023-11-19
> > **Authors Response (Part 2)**
> >
> > Furthermore, we would like to address your questions below:
> >
> > > 1. Could the author include the line of work in constraint active search [2][3] in a discussion of related work? They are closely related to the CBO problem as it aims at searching feasible points efficiently within feasible regions defined by unknown constraints.
> >
> > We will include the line of work in constraint active search [2,3] together with the level set estimation [1] in the related works. They are related to constrained BO since these approaches explore the feasible regions (i.e., either the boundary in the level set estimation [1] or the interior in the constraint active search [2,3]). It is also noteworthy that these approaches differ from constrained BO by not taking into account an objective function.
> >
> >
> > > 2. There is a recent paper studying a similar CBO solution [4] to the proposed algorithm. It is unnecessary to include it in the paper due to its release publication time, but I encourage the author to explore it.
> >
> > We appreciate your update regarding the recent paper [4]. It appears to have been conducted concurrently with our research. The paper seems to investigate a different notion of regret and employ a different BO strategy. We plan to delve into the paper more thoroughly for a detailed comparison.
> >
> > We sincerely hope that our discussion on the related works and the definition of regrets will foster your positive opinion of our work.

---

> > > ### Author Response · Authors · 2023-11-23
> > > **Thank You for Your Feedback and Evaluation**
> > >
> > > As we are approaching the end of the discussion phase, we appreciate the time you dedicated to evaluating our paper and assisting us in identifying works related to CBO in constraint active search and active learning for level set estimation. Thank you for highlighting that our regret can effectively tackle the challenge of infinite instantaneous regret when querying points outside the feasible region.

---

### Meta-Review · Area_Chair_cciU · 2023-12-06

**Metareview:**

This paper considers Bayesian optimization in the decoupled constrained setting (e.g., when the objective function and constraint function(s) are evaluated independently) and achieves novel (albeit fairly standard) regret bounds in this setting. Ultimately, the reviewers and I agree there's a pretty reasonable contribution here. The decoupled setting is indeed pretty understudied in Bayesian optimization, and many acquisition function strategies simply don't work outright in the worst case or fail to take advantage of the decoupling in the best case. The authors method is pretty simple, and I'd like to have seen experimental evaluation on significantly more challenging objective functions, but going through the algebra to pull a typical GP-UCB-like regret bound at the end is reasonably nice.

**Justification For Why Not Higher Score:**

The authors method is pretty simple, and I'd like to have seen experimental evaluation on significantly more challenging objective functions.

**Justification For Why Not Lower Score:**

The setting is pretty understudied, so while I'd have expected to see a vastly more robust experimental results section in a typical constrained BO paper, I think the decoupled setting is different enough that it's mostly fine.

---

### Decision · Program_Chairs · 2024-01-16

Accept (poster)